# Geometric Characterisation and Structured Trajectory Surrogates for Clinical Dataset Condensation[*]

## Abstract

Dataset condensation constructs compact synthetic datasets that retain the training utility of large real-world datasets, enabling efficient model development and potentially supporting downstream research in governed domains such as healthcare. Trajectory matching (TM) is a widely used condensation approach that supervises synthetic data using changes in model parameters observed during training on real data, yet the structure of this supervision signal remains poorly understood. In this paper, we provide a geometric characterisation of trajectory matching, showing that a fixed synthetic dataset can only reproduce a limited span of such training-induced parameter changes. When the resulting supervision signal is spectrally broad, this creates a conditional representability bottleneck. Motivated by this mismatch, we propose Bézier Trajectory Matching (BTM), which replaces SGD trajectories with quadratic Bézier trajectory surrogates between initial and final model states. These surrogates are optimised to reduce average loss along the path while replacing broad SGD-derived supervision with a more structured, lower-rank signal that is better aligned with the optimisation constraints of a fixed synthetic dataset, and they substantially reduce trajectory storage. Experiments on five clinical datasets demonstrate that BTM consistently matches or improves upon standard trajectory matching, with the largest gains in low-prevalence and low-synthetic-budget settings. These results indicate that effective trajectory matching depends on structuring the supervision signal rather than reproducing stochastic optimisation paths.

## 1 INTRODUCTION

Modern machine learning systems increasingly rely on large-scale datasets (Pastorino et al., 2019), yet storing, sharing, and repeatedly training on such datasets remains computationally and operationally expensive. In clinical settings, this challenge is compounded by strict governance and access constraints on electronic health records (EHRs) (Thakur et al., 2024; Shabani, 2019), which limit data sharing, restrict multi-centre collaboration, and can make external validation across diverse real-world cohorts more difficult. Dataset condensation (Wang et al., 2018; Zhao & Bilen, 2021a) addresses this problem by constructing compact synthetic datasets that retain the training utility of the original data. In governed domains, such condensed datasets may support downstream methodological research and model development where direct data access is restricted. However, they do not by themselves provide formal privacy guarantees, and their safe deployment would require combining them with differential privacy.

Trajectory Matching (TM) (Cazenavette et al., 2022; Du et al., 2023; Guo et al., 2024) is a widely used approach to dataset condensation. Rather than matching gradients or feature statistics at a single point, TM supervises synthetic data using changes in model parameters observed during training on real data. By transferring optimisation dynamics into synthetic datasets, TM provides a direct mechanism for encoding task-relevant structure. Despite strong empirical performance, however, progress in TM has largely relied on empirical heuristics—such as trajectory regularisation, selecting which parts of the training trajectory to match, smoothing (Du et al., 2023; Guo et al., 2024; Zhong et al., 2025), and scalability improvements (Cui et al., 2023)—without a principled understanding of the supervision signal being matched.

---

[*]An earlier version of this work appeared as a preprint.

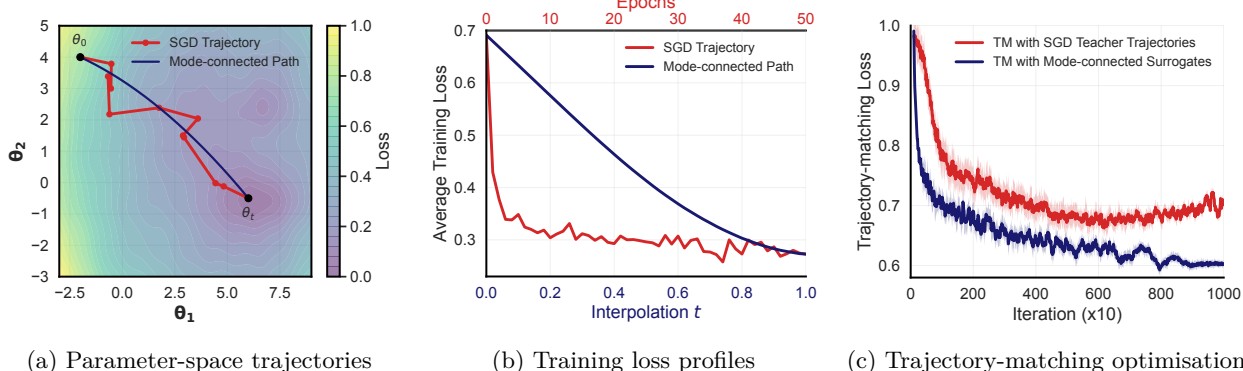

(a) Parameter-space trajectories    (b) Training loss profiles    (c) Trajectory-matching optimisation

Figure 1: Illustrative comparison between raw SGD teacher trajectories and Bézier trajectory surrogates used in BTM. (a) Between the same initial and final model states, raw SGD can follow jagged, directionally variable paths, whereas a Bézier surrogate (mode connected path) provides a smoother low-complexity connection. (b) The two forms of supervision also differ functionally. SGD training exhibits irregular epoch-to-epoch loss behaviour, while the Bézier surrogate defines a smooth loss profile between the same endpoints. (c) In trajectory matching, replacing raw SGD segments with structured surrogate supervision can yield a more stable optimisation objective and faster convergence.

This leaves a basic question unresolved. Which aspects of real-data training trajectories can a fixed synthetic dataset actually reproduce? In TM, supervision is conveyed through changes in model parameters observed during training on real data, but the structure of this signal and its representational demands are not well understood. Without such an understanding, it remains unclear why trajectory matching performs well in some settings, degrades in others, or how its supervision should be structured to make condensation more effective.

In this work, we answer this question through a geometric characterisation of trajectory matching. We show that, for any fixed synthetic dataset, the trajectory matching residual is lower bounded by how well the parameter changes induced by training on real data can be approximated within the gradient span reachable from that dataset. This leads to a conditional representability bottleneck. When the resulting supervision signal is spectrally broad relative to the reachable span, some components necessarily lie outside what the synthetic dataset can reproduce, and a non-zero matching error remains unavoidable.

We then study this mismatch empirically for the supervision signal induced by standard SGD training. We find that the parameter changes used by TM often exhibit broad spectral structure, with non-negligible mass distributed across many weakly aligned directions. This effect appears across stochastic mini-batching, random initialisations, and distinct trained solutions, although we do not attempt to isolate the contribution of each source. The mismatch is especially pronounced in low-prevalence settings, where updates associated with the minority class are sparser and less consistently reinforced during training. As a result, standard TM may require a fixed synthetic dataset to reproduce a supervision signal that is broader than its reachable span, offering a plausible explanation for weaker performance in rare-event clinical prediction.

To mitigate this limitation, we propose *Bézier Trajectory Matching* (BTM), which replaces discrete SGD trajectories with task-optimised quadratic Bézier surrogates connecting the initial and final model states. The approach is inspired by mode connectivity (Garipov et al., 2018; Draxler et al., 2018; Thakur et al., 2025), but our setting differs from the classical case because one endpoint is a random initialisation rather than a second converged solution. A quadratic Bézier curve provides a simple nonlinear path between the endpoints through a single learned control point, allowing the surrogate to capture curvature without storing a long sequence of SGD iterates. In practice, this yields smoother start-to-solution paths whose loss decreases more consistently from initialisation to solution. Compared with standard SGD trajectories, the resulting supervision signal is more structured and has lower effective rank, making it better aligned with the representational constraints of a fixed synthetic dataset while also substantially reducing trajectory storage. Figure 1 provides an intuitive comparison between raw SGD supervision and the structured Bézier surrogates used in BTM. The figure

illustrates the smoother parameter-space geometry and loss profile of the surrogate paths, together with their effect on trajectory-matching optimisation.

We evaluate BTM on five real-world clinical datasets spanning tabular and time-series modalities. Across these tasks, BTM matches or outperforms prior baselines, with the largest gains in low-prevalence and low-synthetic-budget settings, where identifying positive cases reliably is especially important.

The main contributions of this paper are summarised as follows:

- GEOMETRIC CHARACTERISATION OF TRAJECTORY MATCHING: We formulate TM for a fixed synthetic dataset as a constrained subspace-approximation problem and show that its residual is lower bounded by the projection error of real-data parameter changes onto the dataset's reachable gradient span. This reveals a conditional representability bottleneck when that span is limited.

- EMPIRICAL ANALYSIS OF STANDARD TRAJECTORY SUPERVISION: We show that supervision induced by standard SGD trajectories is often spectrally broad, especially in low-prevalence settings, helping to explain the mismatch between real-data training dynamics and what compact synthetic datasets can reproduce.

- BÉZIER TRAJECTORY MATCHING: We introduce BTM, which replaces stored SGD trajectories with task-optimised quadratic Bézier surrogates, yielding a more structured supervision signal and a substantially more compact trajectory representation.

The remainder of the paper is organised as follows. Section 2 reviews related work, Section 3 introduces background on dataset condensation and trajectory matching, Section 4 develops the geometric analysis, Section 5 introduces BTM, Section 6 describes the experimental setup, and Section 7 presents the empirical results.

## 2 RELATED WORK

**Dataset Condensation:** Dataset condensation methods differ primarily in the training signal used to optimise synthetic data. Gradient-based approaches match gradients induced by real and synthetic batches, with later extensions incorporating augmentation-aware objectives (Zhao & Bilen, 2021a;b). Feature- and distribution-based methods instead align representations of real and synthetic data in embedding space, for example through feature alignment or distribution matching (Zhao & Bilen, 2023; Wang et al., 2022). Kernel-based approaches provide more principled closed-form or function-space objectives, but are often less scalable to larger datasets and architectures (Nguyen et al., 2021; 2022). Other scalable or decoupled approaches prioritise efficiency, realism, or data reconstruction rather than explicitly modelling optimisation dynamics (Kanagavelu et al., 2024; Yin et al., 2023; Sun et al., 2023).

**Trajectory-Based Condensation:** Trajectory matching (Cazenavette et al., 2022) is particularly relevant to our work because it supervises synthetic data using parameter displacements observed along real training trajectories. Subsequent work has improved this paradigm through flatter or regularised trajectories (Du et al., 2023), stage- or difficulty-aware selection of which parts of the trajectory to match (Guo et al., 2024), and memory-efficient scaling to larger datasets such as ImageNet (Cui et al., 2023). More recently, Zhong et al. (Zhong et al., 2025) proposed matching convexified trajectories to improve stability, convergence speed, and storage efficiency. Their convexified teacher trajectory provides smoother guidance, but it is not explicitly designed to remain in a low-loss region of the parameter space. As a result, in highly non-convex landscapes, smoothing by convexification may trade trajectory stability against fidelity to the original optimisation dynamics.

**Mode Connectivity:** Mode connectivity studies low-loss curves between trained models (Garipov et al., 2018; Draxler et al., 2018), revealing substantial geometric structure in neural-network parameter space. Related work has shown that these curves can be exploited for ensembling and wider optima (Izmailov et al., 2018), and that their optimisation can benefit from explicitly accounting for weight-space symmetries (Tatro et al., 2020). Recent work has also used related mode-connectivity ideas in clinical machine learning settings (Thakur et al., 2025). Our setting differs from the classical mode-connectivity case because one endpoint is a random initialisation rather than a second converged solution. We therefore use the same geometric

machinery to learn a smoother descent surrogate, rather than to claim a classical low-loss connection between two minima.

**Positioning of Our Work:** This paper sits at the intersection of trajectory-based condensation and geometric modelling of weight-space curves. Rather than proposing another heuristic modification of stored SGD trajectories, we study the representational structure of the supervision signal induced by trajectory matching. This yields a geometric account of when trajectory supervision is limited by the reachable gradient span of a fixed synthetic dataset, and motivates Bézier trajectory surrogates as a more structured alternative to raw SGD trajectories. In contrast to prior trajectory-based methods, including recent approaches based on convexified teacher trajectories (Zhong et al., 2025), our work asks which aspects of trajectory-induced supervision are reproducible by a fixed synthetic dataset. Whereas convexified trajectories primarily smooth teacher dynamics for improved stability and storage efficiency, BTM replaces each teacher trajectory with a task-optimised quadratic Bézier surrogate motivated by this representability bottleneck, yielding a supervision signal that is more structured and better aligned with what a fixed synthetic dataset can reproduce.

## 3 Background

### 3.1 Dataset Condensation Problem

Dataset condensation aims to represent the task-relevant training signal of a large dataset $\mathcal{D}$ with a much smaller synthetic dataset $\tilde{\mathcal{D}}$, such that models trained on $\tilde{\mathcal{D}}$ achieve comparable generalisation performance.

Formally, let

$$\mathcal{D} = \{(\mathbf{x}_i, \mathbf{y}_i)\}_{i=1}^{|\mathcal{D}|}, \qquad \tilde{\mathcal{D}} = \{(\tilde{\mathbf{x}}_j, \tilde{\mathbf{y}}_j)\}_{j=1}^{|\tilde{\mathcal{D}}|}, \qquad |\tilde{\mathcal{D}}| \ll |\mathcal{D}|,$$

and let $\boldsymbol{\theta} \in \Theta = \mathbb{R}^p$ denote model parameters. Given an initialisation $\hat{\boldsymbol{\theta}}_0 \sim P$ and a fixed training procedure $\hat{\boldsymbol{\theta}}_N = \text{Train}(\hat{\boldsymbol{\theta}}_0, \tilde{\mathcal{D}})$, dataset condensation seeks a synthetic dataset satisfying

$$\tilde{\mathcal{D}}^\star \in \arg\min_{\tilde{\mathcal{D}}} \mathbb{E}_{\hat{\boldsymbol{\theta}}_0 \sim P}\big[\mathcal{L}_{\text{val}}(\hat{\boldsymbol{\theta}}_N)\big] \quad \text{s.t.} \quad \hat{\boldsymbol{\theta}}_N = \text{Train}(\hat{\boldsymbol{\theta}}_0, \tilde{\mathcal{D}}), \tag{1}$$

where $\mathcal{L}_{\text{val}}$ denotes a validation loss. The challenge is that $\tilde{\mathcal{D}}$ must induce training dynamics that generalise well despite being orders of magnitude smaller than $\mathcal{D}$.

### 3.2 Trajectory Matching

A prominent approach to dataset condensation is trajectory matching, which uses a teacher optimisation trajectory as supervision for learning synthetic data. Let $\{\boldsymbol{\theta}_\tau\}_{\tau=0}^T \subset \Theta$ denote the parameter sequence obtained by training a teacher model on $\mathcal{D}$. We consider a collection of index pairs $\{(s_k, e_k)\}_{k=1}^K$, with $0 \leq s_k < e_k \leq T$, defining the endpoints of $K$ trajectory segments. Each segment induces a teacher displacement

$$\boldsymbol{\Delta}_k := \boldsymbol{\theta}_{e_k} - \boldsymbol{\theta}_{s_k}.$$

To match segment $k$, a student model is initialised at $\hat{\boldsymbol{\theta}}_0 := \boldsymbol{\theta}_{s_k}$ and trained on $\tilde{\mathcal{D}}$ for $N$ inner steps. Under the first-order, no-momentum update used in standard trajectory matching implementations, the student evolves as

$$\hat{\boldsymbol{\theta}}_{n+1} = \hat{\boldsymbol{\theta}}_n - \eta_s \, \mathbf{g}_n, \qquad \mathbf{g}_n = \nabla_{\boldsymbol{\theta}} \ell(\hat{\boldsymbol{\theta}}_n; B_n), \quad n = 0, \dots, N-1, \tag{2}$$

where $\eta_s$ is the student learning rate, $B_n \subset \tilde{\mathcal{D}}$ is a synthetic mini-batch, and $\ell$ denotes the mini-batch loss. This yields the student displacement

$$\boldsymbol{\delta}_k := \hat{\boldsymbol{\theta}}_N - \boldsymbol{\theta}_{s_k}.$$

The standard per-segment trajectory matching objective is

$$\mathcal{L}_{\text{TM}}(s_k, e_k; \tilde{\mathcal{D}}) := \left\| \hat{\boldsymbol{\theta}}_N - \boldsymbol{\theta}_{e_k} \right\|_2^2 = \|\boldsymbol{\delta}_k - \boldsymbol{\Delta}_k\|_2^2. \tag{3}$$

Trajectory matching can therefore be viewed as learning a synthetic dataset whose induced optimisation steps reproduce teacher displacements in parameter space. In the next section, we introduce our geometric analysis of this objective and show that the student can only realise displacements lying in a restricted span determined by the gradients accessible from the synthetic data.

## 4   A Geometric View of Trajectory Matching

This section develops a geometric analysis of trajectory matching. For a fixed synthetic dataset and inner-loop optimiser, the student cannot realise arbitrary teacher displacements; it can move only along directions generated by gradients accessible from $\tilde{\mathcal{D}}$. This induces a constrained subspace of parameter space that determines which teacher updates are representable.

Two consequences follow. For an individual segment, the trajectory matching loss is lower bounded by the component of the teacher displacement lying outside the student's reachable span. Across segments, this leads to a rank-based bottleneck showing that broad-spectrum teacher supervision cannot, in general, be reproduced within a low-dimensional reachable span. These results explain why raw SGD trajectories can be difficult to match with compact synthetic datasets and motivate structured surrogate supervision.

### 4.1   Reachable Gradient Span

To formalise the representational constraint imposed by the synthetic inner loop, we define the reachable gradient span.

**Definition 1** (Reachable gradient span). *Let $\{\mathbf{g}_n\}_{n=0}^{N-1}$ be the gradients in equation 2 for segment $k$. The reachable gradient span for segment $k$ is*

$$\mathcal{G}_k = \mathrm{span}\{\mathbf{g}_0, \ldots, \mathbf{g}_{N-1}\} \subset \mathbb{R}^p. \tag{4}$$

Under the first-order, no-momentum update in equation 2, the student displacement satisfies

$$\boldsymbol{\delta}_k = -\eta_s \sum_{n=0}^{N-1} \mathbf{g}_n,$$

and therefore $\boldsymbol{\delta}_k \in \mathcal{G}_k$. For a fixed segment, trajectory matching is therefore not an unconstrained regression problem in parameter space. The student can approximate the teacher displacement $\boldsymbol{\Delta}_k$ only by vectors lying in the subspace $\mathcal{G}_k$.

### 4.2   Projection Lower Bound

The subspace constraint above leads directly to a geometric lower bound on the trajectory matching objective.

**Theorem 1** (Projection lower bound). *Fix a segment $k$ with endpoint indices $(s_k, e_k)$, let $P_{\mathcal{G}_k}$ denote the orthogonal projector onto $\mathcal{G}_k$, and let $I$ denote the identity operator on $\mathbb{R}^p$. Then*

$$\mathcal{L}_{\mathrm{TM}}(s_k, e_k; \tilde{\mathcal{D}}) \ \geq \ \left\| (I - P_{\mathcal{G}_k})\boldsymbol{\Delta}_k \right\|_2^2. \tag{5}$$

*Moreover, the minimiser of*

$$\min_{\mathbf{v} \in \mathcal{G}_k} \|\mathbf{v} - \boldsymbol{\Delta}_k\|_2^2$$

*is $P_{\mathcal{G}_k}\boldsymbol{\Delta}_k$. Thus, the bound is tight as a best-in-subspace approximation statement, although a fixed synthetic dataset need not realise this minimiser.*

*Proof.* Section B.1 of the appendix provides the complete proof. $\qquad\square$

**Remark 1** (Best-in-span versus realised displacement)**.** *Theorem 1 characterises the best approximation to* $\boldsymbol{\Delta}_k$ *available within the reachable span* $\mathcal{G}_k$. *The realised student displacement*

$$\boldsymbol{\delta}_k = -\eta_s \sum_{n=0}^{N-1} \mathbf{g}_n$$

*need not coincide with this optimum for a fixed synthetic dataset. The result should therefore be interpreted as a geometric lower bound rather than as an attainability guarantee.*

Theorem 1 shows that any component of the teacher displacement orthogonal to the student's reachable span produces irreducible approximation error. Mismatch is therefore unavoidable whenever the teacher supervision places substantial mass outside the directions that the synthetic inner loop can generate.

### 4.3 A Rank Bottleneck for Shared Synthetic Supervision

The single-segment view above extends naturally to the full collection of teacher displacements supervised by a shared synthetic dataset. Let

$$\mathbf{A} := [\boldsymbol{\Delta}_1, \ldots, \boldsymbol{\Delta}_K] \in \mathbb{R}^{p \times K}, \qquad r_{\mathbf{A}} := \mathrm{rank}(\mathbf{A}), \tag{6}$$

denote the matrix of teacher displacements, and define the global reachable span induced by $\tilde{\mathcal{D}}$ as

$$\mathcal{G} := \mathrm{span}\{\mathbf{g}_n^{(k)} : k = 1, \ldots, K, \ n = 0, \ldots, N-1\}, \qquad r := \dim(\mathcal{G}). \tag{7}$$

For a fixed synthetic dataset, trajectory matching can therefore be viewed as approximating the displacement matrix $\mathbf{A}$ within an $r$-dimensional subspace $\mathcal{G}$. This yields a global representability bottleneck.

**Theorem 2** (Rank bottleneck under an $r$-dimensional reachable span)**.** *Let* $\mathbf{A} = [\boldsymbol{\Delta}_1, \ldots, \boldsymbol{\Delta}_K] \in \mathbb{R}^{p \times K}$ *be the teacher displacement matrix, let* $\mathcal{G} \subset \mathbb{R}^p$ *be a subspace of dimension* $r$ *with orthogonal projector* $P_{\mathcal{G}}$, *let* $I$ *denote the identity operator on* $\mathbb{R}^p$, *and define* $\mathcal{L}_{\mathrm{TM}}^{(k)} := \mathcal{L}_{\mathrm{TM}}(s_k, e_k; \tilde{\mathcal{D}})$. *Then*

$$\frac{1}{K} \sum_{k=1}^{K} \mathcal{L}_{\mathrm{TM}}^{(k)} \ \geq \ \frac{1}{K} \big\|(I - P_{\mathcal{G}})\mathbf{A}\big\|_F^2. \tag{8}$$

*Moreover, among all* $r$-*dimensional subspaces* $\mathcal{G} \subset \mathbb{R}^p$, *the minimum residual is*

$$\min_{\dim(\mathcal{G})=r} \frac{1}{K} \big\|(I - P_{\mathcal{G}})\mathbf{A}\big\|_F^2 = \frac{1}{K} \sum_{j=r+1}^{r_{\mathbf{A}}} \sigma_j(\mathbf{A})^2, \tag{9}$$

*where* $\sigma_j(\mathbf{A})$ *denotes the* $j$-*th singular value of* $\mathbf{A}$.

*Proof.* Section B.2 of the appendix provides the complete proof. $\square$

**Remark 2.** *The displacements* $\{\boldsymbol{\Delta}_k\}_{k=1}^{K}$ *may be drawn from one or multiple teacher trajectories (Cazenavette et al., 2022). The analysis depends only on the resulting collection of displacement vectors and applies in either case.*

**Remark 3** (Conditional nature of the bottleneck)**.** *Theorem 2 is conditional on the effective dimension* $r = \dim(\mathcal{G})$ *induced by a fixed synthetic dataset, inner-loop horizon, and model class. It does not assert a universal, architecture-independent upper bound on* $r$. *Instead, it quantifies the residual once an* $r$-*dimensional reachable span is given.*

Theorem 2 identifies a fundamental limitation of trajectory matching under constrained synthetic supervision. When substantial spectral mass of $\mathbf{A}$ lies outside the effective reachable span, a positive residual remains unavoidable. The difficulty of trajectory matching is therefore governed not only by the nominal rank of $\mathbf{A}$, but more generally by its spectrum and effective rank relative to the student's reachable span.

These results suggest that effective trajectory supervision should be concentrated in a small number of coherent directions. The next section introduces Bézier Trajectory Matching, which is designed to impose exactly this kind of structured supervision.

## 5 Bézier Trajectory Matching

The rank bottleneck above suggests that effective supervision should preserve functionally meaningful progress while avoiding diffuse high-rank variability. *Bézier Trajectory Matching* (BTM) addresses this limitation by replacing stochastic optimisation trajectories with learnable quadratic Bézier curves that act as structured low-rank surrogates. These surrogate paths are optimised to reduce average loss along the path, suppress unnecessary high-loss excursions, and empirically encourage a smoother, typically decreasing expected loss profile from initialisation to solution. The induced displacement directions are therefore confined to a low-dimensional subspace, yielding supervision that is better aligned with the constraints of student optimisation.

The construction of the surrogate trajectories is described first. Their displacement structure and the resulting supervision are then analysed.

### 5.1 Bézier Surrogate Trajectories

Surrogate trajectories are constructed between model initialisations and their corresponding trained solutions using quadratic Bézier curves. Let $\{(\boldsymbol{\theta}_0^{(m)}, \boldsymbol{\theta}_T^{(m)})\}_{m=1}^M$ denote initialisation–solution pairs obtained from real-data SGD trajectories. For each pair, define a quadratic Bézier trajectory

$$\Phi^{(m)}(t) = (1-t)^2 \boldsymbol{\theta}_0^{(m)} + 2t(1-t)\boldsymbol{\phi}^{(m)} + t^2 \boldsymbol{\theta}_T^{(m)}, \qquad t \in [0,1], \tag{10}$$

where $\boldsymbol{\phi}^{(m)} \in \mathbb{R}^p$ is a learnable control point. The straight line is recovered by the special choice

$$\boldsymbol{\phi}^{(m)} = \tfrac{1}{2}\big(\boldsymbol{\theta}_0^{(m)} + \boldsymbol{\theta}_T^{(m)}\big),$$

so the quadratic family strictly contains the linear interpolation baseline.

Each control point is optimised to reduce the average loss along the path,

$$\boldsymbol{\phi}^{(m)\star} \in \arg\min_{\boldsymbol{\phi}} \int_0^1 \mathcal{L}\big(\Phi_{\boldsymbol{\phi}}^{(m)}(t); \mathcal{D}\big) \, dt, \tag{11}$$

where $\mathcal{L}(\boldsymbol{\theta}; \mathcal{D})$ denotes the dataset-level training loss evaluated on $\mathcal{D}$.

In practice, the integral is approximated using discrete samples of $t \in [0,1]$. This optimisation is related to mode connectivity (Garipov et al., 2018), but differs in an important respect. Because one endpoint is the random initialisation, the curve is not interpreted as a classical mode connection between two low-loss modes. Instead, minimising the average loss along the path encourages a smoother curve with lower average loss and, empirically, a typically decreasing expected loss profile from initialisation to solution.

Each surrogate trajectory is specified by three parameter vectors, $\boldsymbol{\theta}_0^{(m)}$, $\boldsymbol{\phi}^{(m)\star}$, and $\boldsymbol{\theta}_T^{(m)}$. Storing $M$ surrogates therefore requires $\mathcal{O}(3M)$ parameter vectors. Standard trajectory matching, by contrast, stores $T$ checkpoints per trajectory and therefore requires $\mathcal{O}(TM)$ parameter vectors. The surrogate representation thus yields a substantial reduction in memory when $T \gg 1$.

### 5.2 Low-Rank Supervision from Bézier Surrogates

The surrogate trajectories above induce the displacement supervision used in trajectory matching. For a surrogate trajectory $\Phi^{(m)}$, define segment displacements

$$\boldsymbol{\Delta}^{(m)}(t_s, t_e) = \Phi^{(m)}(t_e) - \Phi^{(m)}(t_s), \qquad 0 \le t_s < t_e \le 1. \tag{12}$$

Collecting such displacements across sampled segments yields a surrogate displacement matrix analogous to $\mathbf{A}$ in Section 4. The key structural property is that all segment displacements from a single quadratic Bézier trajectory lie in a two-dimensional subspace. For notational simplicity, the surrogate index $m$ is suppressed in the theorem statement.

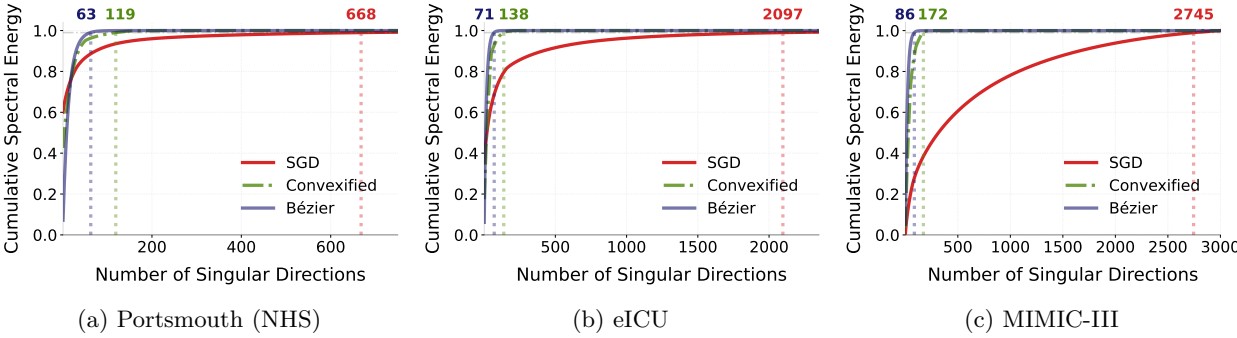

(a) Portsmouth (NHS)        (b) eICU        (c) MIMIC-III

Figure 2: Effective dimensionality of teacher displacement supervision across datasets. Each panel shows the cumulative spectral energy of the displacement matrix as a function of the number of singular directions, computed from 50 teacher trajectories per method. SGD supervision exhibits high effective rank, with energy spread across many directions. In contrast, Bézier surrogates concentrate spectral mass in a few directions, while convexified trajectories are intermediate.

**Theorem 3** (Low-dimensional displacement structure of quadratic Bézier surrogates)**.** *Let $\Phi(t)$ be a quadratic Bézier trajectory as defined in equation [10]. Then for any $0 \leq t_s < t_e \leq 1$,*

$$\Phi(t_e) - \Phi(t_s) = (t_e - t_s)(\boldsymbol{\theta}_T - \boldsymbol{\theta}_0) + (t_e - t_s)(1 - t_s - t_e)\big(2\boldsymbol{\phi} - \boldsymbol{\theta}_0 - \boldsymbol{\theta}_T\big). \tag{13}$$

*Consequently, every segment displacement lies in the subspace*

$$\mathrm{span}\Big\{\boldsymbol{\theta}_T - \boldsymbol{\theta}_0,\ 2\boldsymbol{\phi} - \boldsymbol{\theta}_0 - \boldsymbol{\theta}_T\Big\},$$

*and therefore*

$$V_\Phi := \mathrm{span}\big\{\Phi(t_e) - \Phi(t_s):\ 0 \leq t_s < t_e \leq 1\big\}$$

*satisfies $\dim(V_\Phi) \leq 2$. In particular, any displacement matrix constructed from segments of a single surrogate trajectory has rank at most $2$.*

*Proof.* Section [B.3] of the appendix provides the complete proof. $\square$

For $M$ surrogate trajectories, the combined supervision therefore has nominal rank at most $2M$. Empirically, however, the effective rank is substantially lower and typically scales closer to $M$ than to $2M$ (Figure [2]). BTM therefore replaces diffuse high-rank supervision with a controlled low-rank alternative that is better aligned with the constraints of student optimisation. Rather than matching a broad spectrum of weakly aligned directions, the synthetic dataset needs only to reproduce a small number of coherent displacement directions that capture the dominant structure of optimisation.

### 5.3 Spectral Structure of Trajectory Supervision

The geometric analysis in Section [4] shows that the difficulty of trajectory matching depends not only on the magnitude of teacher displacements, but also on how their spectral energy is distributed. By Theorem [2], for any fixed effective reachable dimension $r$, slower spectral concentration of the displacement matrix $\mathbf{A}$ implies larger tail energy beyond its first $r$ singular directions and therefore a stronger conditional representability bottleneck. The low-dimensional displacement structure established above suggests that Bézier surrogates should induce supervision that is more spectrally concentrated than raw SGD trajectories.

Figure [2] compares the cumulative spectral energy of displacement matrices computed from 50 teacher trajectories per method across datasets. Raw SGD supervision exhibits slow spectral concentration, with energy spread across many weakly aligned directions. Convexified trajectories are more concentrated but remain relatively diffuse, whereas Bézier surrogates concentrate most of their spectral mass in far fewer directions. By Theorem [2], faster spectral concentration means that a larger fraction of the supervision

signal can be captured within any fixed reachable span of dimension $r$, leaving a smaller residual tail and therefore a weaker conditional representability bottleneck. This comparison should therefore be interpreted as a teacher-side diagnostic of the bottleneck, since it characterises the spectrum of $\mathbf{A}$ in practice rather than the student's reachable span itself.

This contrast in spectral concentration is especially relevant in clinical settings, where low-prevalence outcomes often give rise to sparse and inconsistently aligned minority-class updates. These can introduce low-energy but task-relevant components into the supervision signal, making faithful reproduction more difficult for a compact synthetic dataset. The stronger concentration observed for Bézier surrogates is therefore consistent with the design goal of BTM, namely to preserve dominant, functionally meaningful progress while suppressing diffuse high-rank variability that is harder for a fixed synthetic dataset to reproduce.

### 5.4 Bézier Trajectory Matching Objective

BTM replaces the teacher displacements $\boldsymbol{\Delta}_k$ in trajectory matching with segment displacements sampled from learned Bézier surrogates, as defined in equation 12. In all other respects, the training pipeline follows standard trajectory matching.

Assume access to a collection of $M$ optimised Bézier surrogates $\{\Phi^{(m)}\}_{m=1}^{M}$ constructed from real-data training, together with a synthetic dataset $\tilde{\mathcal{D}} = \{(\tilde{\mathbf{x}}_i, \tilde{\mathbf{y}}_i)\}_{i=1}^{|\tilde{\mathcal{D}}|}$. The synthetic inputs are stacked into a matrix $\tilde{\mathbf{X}} \in \mathbb{R}^{|\tilde{\mathcal{D}}| \times d_x}$, and the corresponding labels into a tensor $\tilde{\mathbf{Y}} \in \mathbb{R}^{|\tilde{\mathcal{D}}| \times C}$.

At each outer iteration, a surrogate index $m \sim \mathcal{U}\{1, \ldots, M\}$ and time parameters $0 \le t_s < t_e \le 1$ are sampled, and the corresponding segment endpoints are set to

$$\boldsymbol{\theta}_s = \Phi^{(m)}(t_s), \qquad \boldsymbol{\theta}_e = \Phi^{(m)}(t_e). \tag{14}$$

A student model is initialised at $\hat{\boldsymbol{\theta}}_0 = \boldsymbol{\theta}_s$ and trained for $N$ steps using the update in equation 2, producing $\hat{\boldsymbol{\theta}}_N$. In practice, optimisation uses the normalised trajectory matching loss

$$\mathcal{L}_{\mathrm{BTM}} = \frac{\|\hat{\boldsymbol{\theta}}_N - \boldsymbol{\theta}_e\|_2^2}{\|\boldsymbol{\theta}_s - \boldsymbol{\theta}_e\|_2^2}. \tag{15}$$

For a fixed sampled segment, the denominator depends only on the surrogate endpoints and is therefore a positive constant with respect to the student trajectory. Consequently, the student displacement that minimises equation 15 is the same as the one that minimises the unnormalised residual in equation 3, so the representability analysis in Section 4 still applies at the segment level. The effect of the normalisation is to reduce scale variation across segments, making short and long segments more comparable, while reweighting them by inverse squared segment length. It should therefore not be interpreted as a purely directional objective, but rather as a stabilised implementation of the same underlying segment-level matching problem.

The synthetic dataset is optimised by differentiating $\mathcal{L}_{\mathrm{BTM}}$ through the unrolled student updates. Following standard first-order trajectory matching, the meta-gradient is approximated using the gradient with respect to the final student parameters,

$$\boldsymbol{g}_L = \nabla_{\hat{\boldsymbol{\theta}}_N} \mathcal{L}_{\mathrm{BTM}} = \frac{2(\hat{\boldsymbol{\theta}}_N - \boldsymbol{\theta}_e)}{\|\boldsymbol{\theta}_s - \boldsymbol{\theta}_e\|_2^2}. \tag{16}$$

The corresponding approximate meta-gradient with respect to the synthetic inputs is

$$\nabla_{\tilde{\mathbf{X}}} \mathcal{L}_{\mathrm{BTM}} \approx -\eta_s \sum_{n=0}^{N-1} \frac{1}{|B_n|} \sum_{(\tilde{\mathbf{x}}, \tilde{\mathbf{y}}) \in B_n} \nabla_{\tilde{\mathbf{X}}} \Big\langle \nabla_{\boldsymbol{\theta}} \ell(\hat{\boldsymbol{\theta}}_n; \tilde{\mathbf{x}}, \tilde{\mathbf{y}}), \boldsymbol{g}_L \Big\rangle, \tag{17}$$

where $\eta_s$ is the student learning rate and $B_n$ denotes the synthetic mini-batch at step $n$.

The synthetic inputs are updated by gradient descent,

$$\tilde{\mathbf{X}} \leftarrow \tilde{\mathbf{X}} - \eta_{\mathbf{x}} \nabla_{\tilde{\mathbf{X}}} \mathcal{L}_{\mathrm{BTM}}, \tag{18}$$

and the labels $\tilde{\mathbf{Y}}$ can also be optimised analogously. The complete procedure is given in Algorithm 1 in the appendix.

### 5.5 Functional Justification of Surrogate Supervision

The low-rank analysis above concerns the geometry of the supervision signal. A complementary question is whether the surrogate path remains functionally close to the teacher supervision it replaces. The next result gives a chord-based bound showing that a quadratic Bézier surrogate has controlled curvature and remains close in prediction space to the piecewise-linear interpolation of the teacher checkpoints whenever the surrogate stays near the endpoint chord and the model is parameter-Lipschitz along the relevant paths.

**Theorem 4** (Geometric regularity and prediction fidelity of Bézier surrogates). *Let $\{\boldsymbol{\theta}_\tau\}_{\tau=0}^T \subset \mathbb{R}^p$ denote a teacher trajectory with initialisation $\boldsymbol{\theta}_0$ and endpoint $\boldsymbol{\theta}_T$, and let $\gamma : [0,1] \to \mathbb{R}^p$ denote its piecewise-linear interpolation. Define the endpoint chord*

$$c(t) := (1-t)\boldsymbol{\theta}_0 + t\boldsymbol{\theta}_T,$$

*and let $\Phi(t) := \Phi_{\boldsymbol{\phi}^\star}(t)$ denote the optimised quadratic Bézier surrogate. Define*

$$\kappa := 2\|\boldsymbol{\theta}_0 - 2\boldsymbol{\phi}^\star + \boldsymbol{\theta}_T\|_2, \qquad D := \sup_{t \in [0,1]} \|\gamma(t) - c(t)\|_2.$$

*Assume that for every $\boldsymbol{x} \in \mathcal{X}$, the model map $f_{\boldsymbol{\theta}}(\boldsymbol{x})$ is $L_f$-Lipschitz in $\boldsymbol{\theta}$ on the set $\{\Phi(t), \gamma(t) : t \in [0,1]\}$. Then the following hold.*

  *(i) **Smooth curvature.***

$$\Phi''(t) = 2(\boldsymbol{\theta}_0 - 2\boldsymbol{\phi}^\star + \boldsymbol{\theta}_T), \qquad \sup_{t \in [0,1]} \|\Phi''(t)\|_2 = \kappa. \tag{19}$$

  *Moreover,*

$$\sup_{t \in [0,1]} \|\Phi(t) - c(t)\|_2 = \frac{\kappa}{8}. \tag{20}$$

  *(ii) **Prediction fidelity.***

$$\sup_{\boldsymbol{x} \in \mathcal{X},\, t \in [0,1]} \|f_{\Phi(t)}(\boldsymbol{x}) - f_{\gamma(t)}(\boldsymbol{x})\|_2 \le L_f \left(\frac{\kappa}{8} + D\right). \tag{21}$$

*Proof.* Sections B.4 and B.5 of the appendix provides the complete proof. $\square$

The theorem compares the surrogate $\Phi$ with the piecewise-linear teacher interpolation $\gamma$. The chord $c$ serves only as a geometric reference used to control this deviation. The bound is therefore informative when both the surrogate curvature $\kappa$ and the teacher deviation $D$ are small.

For intuition, let the teacher updates satisfy $\boldsymbol{\theta}_{\tau+1} = \boldsymbol{\theta}_\tau - \eta\mathbf{u}_\tau$, where $\eta > 0$ is a step size and $\mathbf{u}_\tau$ is the update direction at step $\tau$. Define the discrete second-order difference $\mathbf{h}_\tau := \boldsymbol{\theta}_{\tau+1} - 2\boldsymbol{\theta}_\tau + \boldsymbol{\theta}_{\tau-1}$. Then $\mathbf{h}_\tau = -\eta(\mathbf{u}_\tau - \mathbf{u}_{\tau-1})$, so local curvature of the teacher path is governed by step-to-step changes in the update direction. If $\mathbf{u}_\tau = \mathbf{d}_\tau + \boldsymbol{\xi}_\tau$, where $\mathbf{d}_\tau$ denotes systematic drift and $\boldsymbol{\xi}_\tau$ denotes stochastic fluctuation, then

$$\mathbf{h}_\tau = -\eta\big[(\mathbf{d}_\tau - \mathbf{d}_{\tau-1}) + (\boldsymbol{\xi}_\tau - \boldsymbol{\xi}_{\tau-1})\big].$$

Thus, discrete teacher curvature mixes optimisation drift with stochastic variability, whereas the quadratic surrogate has globally smooth curvature.

**Corollary 1** (Conditional supervision sufficiency). *Assume the setting of Theorem 4, and further assume that for every $\boldsymbol{x} \in \mathcal{X}$, the model map $f_{\boldsymbol{\theta}}(\boldsymbol{x})$ is $L_f$-Lipschitz in $\boldsymbol{\theta}$ on the set $\{\gamma_{\text{stu}}(t), \Phi(t), \gamma(t) : t \in [0,1]\}$. Let $\gamma_{\text{stu}} : [0,1] \to \mathbb{R}^p$ denote a student trajectory induced by the synthetic dataset, and suppose that*

$$\sup_{t \in [0,1]} \|\gamma_{\text{stu}}(t) - \Phi(t)\|_2 \le \varepsilon_{\text{syn}}.$$

*Then*

$$\sup_{\boldsymbol{x} \in \mathcal{X},\, t \in [0,1]} \|f_{\gamma_{\text{stu}}(t)}(\boldsymbol{x}) - f_{\gamma(t)}(\boldsymbol{x})\|_2 \le L_f \left(\varepsilon_{\text{syn}} + \frac{\kappa}{8} + D\right). \tag{22}$$

Table 1: Summary of datasets used in this work.

| Source | Dataset | Modality | # Samples | Task |
|---|---|---|---|---|
| UK NHS ED Cohorts | Oxford Portsmouth Birmingham | Tabular Tabular Tabular | 161,955 38,717 95,236 | COVID-19 diagnosis |
| Public ICU Benchmarks | eICU | Tabular | 49,305 | In-hospital mortality |
| | MIMIC-III | Time-series | 21,156 21,728 | In-hospital mortality Phenotyping (25 classes, multi-label) |

*Proof.* Section B.6 of the appendix provides the complete proof. □

Corollary 1 is a conditional sufficiency statement rather than an optimisation guarantee for Algorithm 1. It identifies a condition under which a student trajectory that tracks the surrogate also remains close to the teacher interpolation in prediction space.

Together, Theorem 4 and Corollary 1 show that the surrogate family is not only low-rank, but also functionally meaningful under explicit geometric conditions. Along with the empirical ablations in Section 7.4, this supports BTM as a low-complexity surrogate family that preserves endpoint-defined structure while filtering stepwise variability from teacher supervision.

# 6 Experimental Setup

**Datasets.** Evaluation is conducted on five real-world clinical datasets spanning multiple institutions, modalities, and prediction settings. These include three large de-identified emergency department cohorts collected from UK NHS Trusts in Oxford, Portsmouth, and Birmingham (Soltan et al., 2024), together with two widely used public ICU benchmarks, eICU (Pollard et al., 2018) and MIMIC-III (Johnson et al., 2016). As summarised in Table 1, the resulting benchmark suite covers both tabular and multivariate time-series data, and includes binary and multi-label classification tasks. This combination enables evaluation across both institution-specific operational datasets and established public critical-care benchmarks, providing a diverse and clinically realistic test bed for dataset condensation. Additional dataset details are provided in Section C.

Clinical datasets differ from standard vision benchmarks in ways that are directly relevant to trajectory matching. Outcomes are often highly imbalanced, and patient populations are heterogeneous and noisy, leading to optimisation dynamics with sparse and inconsistently aligned gradient signals. As discussed in Section 4.3, this tends to produce diffuse supervision spread across many weakly reinforced directions, which is difficult to capture within the limited reachable span of a compact synthetic dataset. Clinical settings therefore provide a particularly stringent and realistic test bed for trajectory-based condensation.

**Baselines.** The comparison focuses on trajectory-matching-based dataset condensation methods, as these provide the most direct point of reference for BTM. The selected methods differ primarily in how the trajectory supervision signal is constructed or modified.

The baselines include Matching Training Trajectories (MTT) (Cazenavette et al., 2022), which directly matches multi-step parameter updates between real and synthetic training; Flat Trajectory Distillation (FTD) (Du et al., 2023), which promotes flatter trajectories to mitigate accumulated matching error; Difficulty-Aligned Trajectory Matching (DATM) (Guo et al., 2024), which samples trajectory segments from different training stages to vary the difficulty of the supervision signal; and Matching Convexified Trajectories (MCT) (Zhong et al., 2025), which replaces SGD trajectories with convexified surrogates to improve stability and efficiency. TrajEctory matching with Soft Label Assignment (TESLA) (Cui et al., 2023) is not included, as it primarily addresses scalability by reducing memory requirements with respect to the number of unrolled optimisation

steps rather than modifying the structure of the supervision signal itself. Random subset selection is included as a lower-bound baseline, and full-dataset training as an upper bound.

**Experimental Protocol.** Synthetic datasets are constructed at 50, 100, 200, and 500 instances per class (*ipc*). For multi-label phenotyping, we interpret *ipc* as a per-label budget (*plb*) and map it to the total synthetic dataset size as $|\tilde{\mathcal{D}}| = plb \times \frac{L}{\bar{c}}$, where $L$ is the number of labels and $\bar{c}$ is the average label cardinality (i.e., the mean number of positive labels per sample). This ensures that the effective supervision budget is comparable to the single-label setting while accounting for label overlap. All methods are adapted to clinical data modalities following their original implementations, with fixed hard labels used throughout. Across all datasets, 65%, 15%, and 20% of the samples are used for training, validation, and testing, respectively. Following standard evaluation practice, randomly initialised models are trained from scratch on the condensed data and evaluated on the held-out test sets. Performance is measured using the area under the receiver operating characteristic curve (AUROC) and the area under the precision–recall curve (AUPRC). For the multi-label phenotyping, we report *macro*-averaged AUROC and AUPRC across labels. All results are reported over 10 random initialisations.

Teacher trajectories are obtained by training models on the real training data with standard optimisation, with hyperparameters such as the learning rate, optimiser, and training horizon selected on the validation set. The main BTM-specific hyperparameters, including the number of student optimisation steps, the segment length $(t_e - t_s)$, and the optimisation settings for the Bézier control point, are selected in the same way. Full implementation details are provided in Section D of the appendix.

Evaluation architectures are fixed across all methods. A shallow multi-layer perceptron (MLP) (Rumelhart et al., 1986) is used for tabular data, and a temporal convolutional network (TCN) (Bai et al., 2018) for time-series data. Consistent with prior work, the same architecture is used for both condensation and evaluation, except in cross-architecture experiments. Synthetic inputs are initialised from real samples.

## 7 Results and Discussion

### 7.1 Downstream Utility of Condensed Data

The primary evaluation criterion is downstream task performance after training on condensed data. Particular emphasis is placed on low-data and low-prevalence regimes, where diffuse supervision and weakly aligned gradients make trajectory-based condensation especially challenging.

#### 7.1.1 NHS emergency department cohorts.

Table 2 reports results on the three NHS cohorts across synthetic budgets ranging from 50 to 500 instances per class for the task of COVID-19 prediction. BTM performs strongly across all settings and is particularly effective in the clinically relevant low-prevalence regime. The gains are most consistent in AUPRC, suggesting that the surrogate supervision is especially beneficial when positive-class signal is rare and difficult to preserve under severe compression.

On the Portsmouth dataset (5.3% prevalence), BTM achieves the best AUROC at *ipc*=50, 200, and 500, and the best AUPRC from *ipc*=100 onward. In AUPRC, this corresponds to improvements of 2.1%, 5.9%, and 3.2% over the strongest baseline at *ipc*=100, 200, and 500, respectively. At *ipc*=500, BTM reaches 0.609, which is 99.8% of full-dataset performance (0.610). At the smallest budget, DATM is marginally stronger in AUPRC, but BTM overtakes it as the synthetic budget increases.

On the Oxford dataset (1.7% prevalence), BTM again achieves the best AUROC at *ipc*=50, 200, and 500, and attains the strongest AUPRC at every budget. The relative AUPRC gains over the strongest baseline are 11.9%, 2.9%, 3.8%, and 6.0% at *ipc*=50, 100, 200, and 500, respectively. Performance improves steadily from 0.414 at *ipc*=50 to 0.440 at *ipc*=500, corresponding to 98.9% of the full-dataset AUPRC (0.445).

The clearest gains appear on the Birmingham dataset (0.8% prevalence), the most imbalanced cohort. Here, BTM delivers the best AUPRC at every budget and the best AUROC up to *ipc*=200, while remaining competitive at *ipc*=500. The AUPRC gains over the strongest baseline are 15.5%, 17.9%, 16.6%, and 15.0%

Table 2: Performance on the three NHS cohorts across different *ipc* levels for COVID-19 prediction. Best results at each *ipc* are highlighted in blue (ours) and red (baseline).

**(a) Portsmouth dataset** (5.3% prevalence)

| Method | AUROC | | | | AUPRC | | | |
|---|---|---|---|---|---|---|---|---|
| | 50 | 100 | 200 | 500 | 50 | 100 | 200 | 500 |
| RANDOM | $0.835_{\pm 0.006}$ | $0.863_{\pm 0.008}$ | $0.880_{\pm 0.003}$ | $0.877_{\pm 0.008}$ | $0.289_{\pm 0.018}$ | $0.314_{\pm 0.025}$ | $0.388_{\pm 0.016}$ | $0.436_{\pm 0.026}$ |
| MTT | $0.853_{\pm 0.002}$ | $0.868_{\pm 0.001}$ | $0.874_{\pm 0.001}$ | $0.900_{\pm 0.001}$ | $0.518_{\pm 0.001}$ | $0.542_{\pm 0.004}$ | $0.530_{\pm 0.001}$ | $0.526_{\pm 0.001}$ |
| FTD | $0.835_{\pm 0.002}$ | $0.871_{\pm 0.001}$ | $0.883_{\pm 0.001}$ | $0.895_{\pm 0.001}$ | $0.498_{\pm 0.002}$ | $0.532_{\pm 0.002}$ | $0.549_{\pm 0.001}$ | $0.590_{\pm 0.001}$ |
| MCT | $0.867_{\pm 0.001}$ | $0.879_{\pm 0.001}$ | $0.885_{\pm 0.001}$ | $0.893_{\pm 0.001}$ | $0.476_{\pm 0.001}$ | $0.493_{\pm 0.003}$ | $0.508_{\pm 0.001}$ | $0.541_{\pm 0.005}$ |
| DATM | $0.872_{\pm 0.001}$ | $0.884_{\pm 0.001}$ | $0.879_{\pm 0.001}$ | $0.890_{\pm 0.003}$ | $0.561_{\pm 0.001}$ | $0.560_{\pm 0.001}$ | $0.563_{\pm 0.002}$ | $0.580_{\pm 0.001}$ |
| BTM (OURS) | $0.876_{\pm 0.002}$ | $0.880_{\pm 0.002}$ | $0.907_{\pm 0.001}$ | $0.901_{\pm 0.001}$ | $0.557_{\pm 0.001}$ | $0.572_{\pm 0.001}$ | $0.596_{\pm 0.001}$ | $0.609_{\pm 0.001}$ |
| **Full Dataset** | $\mathbf{0.906_{\pm 0.002}}$ | | | | $\mathbf{0.610_{\pm 0.004}}$ | | | |

**(b) Oxford dataset** (1.7% prevalence)

| Method | AUROC | | | | AUPRC | | | |
|---|---|---|---|---|---|---|---|---|
| | 50 | 100 | 200 | 500 | 50 | 100 | 200 | 500 |
| RANDOM | $0.831_{\pm 0.003}$ | $0.856_{\pm 0.004}$ | $0.870_{\pm 0.005}$ | $0.879_{\pm 0.007}$ | $0.149_{\pm 0.002}$ | $0.171_{\pm 0.006}$ | $0.230_{\pm 0.008}$ | $0.256_{\pm 0.007}$ |
| MTT | $0.834_{\pm 0.002}$ | $0.861_{\pm 0.004}$ | $0.855_{\pm 0.008}$ | $0.869_{\pm 0.001}$ | $0.370_{\pm 0.008}$ | $0.391_{\pm 0.006}$ | $0.405_{\pm 0.012}$ | $0.410_{\pm 0.007}$ |
| FTD | $0.851_{\pm 0.008}$ | $0.850_{\pm 0.006}$ | $0.852_{\pm 0.005}$ | $0.874_{\pm 0.006}$ | $0.368_{\pm 0.001}$ | $0.385_{\pm 0.009}$ | $0.400_{\pm 0.009}$ | $0.410_{\pm 0.005}$ |
| MCT | $0.845_{\pm 0.001}$ | $0.862_{\pm 0.003}$ | $0.882_{\pm 0.001}$ | $0.874_{\pm 0.003}$ | $0.341_{\pm 0.014}$ | $0.369_{\pm 0.016}$ | $0.358_{\pm 0.004}$ | $0.352_{\pm 0.002}$ |
| DATM | $0.856_{\pm 0.005}$ | $0.868_{\pm 0.004}$ | $0.875_{\pm 0.003}$ | $0.880_{\pm 0.004}$ | $0.359_{\pm 0.015}$ | $0.413_{\pm 0.005}$ | $0.419_{\pm 0.002}$ | $0.415_{\pm 0.003}$ |
| BTM (OURS) | $0.865_{\pm 0.004}$ | $0.867_{\pm 0.009}$ | $0.886_{\pm 0.005}$ | $0.891_{\pm 0.001}$ | $0.414_{\pm 0.001}$ | $0.425_{\pm 0.002}$ | $0.435_{\pm 0.001}$ | $0.440_{\pm 0.001}$ |
| **Full Dataset** | $\mathbf{0.901_{\pm 0.001}}$ | | | | $\mathbf{0.445_{\pm 0.004}}$ | | | |

**(c) Birmingham dataset** (0.8% prevalence)

| Method | AUROC | | | | AUPRC | | | |
|---|---|---|---|---|---|---|---|---|
| | 50 | 100 | 200 | 500 | 50 | 100 | 200 | 500 |
| RANDOM | $0.842_{\pm 0.011}$ | $0.857_{\pm 0.014}$ | $0.865_{\pm 0.009}$ | $0.890_{\pm 0.12}$ | $0.080_{\pm 0.018}$ | $0.073_{\pm 0.009}$ | $0.119_{\pm 0.016}$ | $0.151_{\pm 0.009}$ |
| MTT | $0.828_{\pm 0.006}$ | $0.847_{\pm 0.013}$ | $0.851_{\pm 0.013}$ | $0.884_{\pm 0.007}$ | $0.189_{\pm 0.027}$ | $0.192_{\pm 0.021}$ | $0.212_{\pm 0.016}$ | $0.234_{\pm 0.017}$ |
| FTD | $0.829_{\pm 0.005}$ | $0.839_{\pm 0.010}$ | $0.847_{\pm 0.010}$ | $0.891_{\pm 0.004}$ | $0.193_{\pm 0.008}$ | $0.218_{\pm 0.013}$ | $0.216_{\pm 0.018}$ | $0.229_{\pm 0.014}$ |
| MCT | $0.860_{\pm 0.001}$ | $0.862_{\pm 0.007}$ | $0.877_{\pm 0.012}$ | $0.886_{\pm 0.001}$ | $0.233_{\pm 0.002}$ | $0.220_{\pm 0.002}$ | $0.234_{\pm 0.003}$ | $0.247_{\pm 0.006}$ |
| DATM | $0.824_{\pm 0.001}$ | $0.832_{\pm 0.022}$ | $0.851_{\pm 0.016}$ | $0.870_{\pm 0.005}$ | $0.178_{\pm 0.004}$ | $0.229_{\pm 0.005}$ | $0.235_{\pm 0.002}$ | $0.240_{\pm 0.006}$ |
| BTM (OURS) | $0.863_{\pm 0.004}$ | $0.872_{\pm 0.004}$ | $0.883_{\pm 0.004}$ | $0.890_{\pm 0.003}$ | $0.269_{\pm 0.007}$ | $0.270_{\pm 0.002}$ | $0.274_{\pm 0.003}$ | $0.284_{\pm 0.002}$ |
| **Full Dataset** | $\mathbf{0.898_{\pm 0.002}}$ | | | | $\mathbf{0.293_{\pm 0.007}}$ | | | |

across the four budgets, highlighting the benefit of BTM in the most data-sparse and low-prevalence setting. At *ipc*=500, BTM reaches 0.284, or 96.9% of full-dataset AUPRC (0.293).

This pattern is consistent with the geometric analysis in Section 4. As prevalence decreases, the supervision signal becomes increasingly diffuse and weakly aligned, making trajectory matching more sensitive to high-rank variability. BTM mitigates this effect by replacing raw SGD supervision with smoother, low-rank surrogate trajectories that retain task-relevant structure while suppressing stepwise noise.

Among the baselines, MCT is generally the most competitive in AUROC, suggesting that reducing supervision rank is already beneficial. Its gains in AUPRC, however, are less consistent, particularly on the lower-prevalence cohorts. This aligns with the methodological difference between the two approaches. MCT constructs convexified trajectory surrogates from stored checkpoints, whereas BTM learns Bézier control points that reduce average path loss and encourage a smoother descent profile. DATM is also competitive on the Portsmouth and Oxford datasets, especially at smaller budgets, but its advantage weakens as prevalence decreases. Overall, these results indicate that learned low-rank surrogate supervision is particularly useful in the low-data, low-prevalence regimes most relevant to clinical practice.

### 7.1.2 Public ICU benchmarks.

Table 3 reports results on the eICU and MIMIC-III datasets across synthetic budgets ranging from 50 to 500 instances per class for the task of in-hospital mortality prediction. As with the NHS cohorts, BTM performs strongly across compression levels, with the clearest and most consistent gains appearing in AUPRC. This

Table 3: Performance on the eICU and MIMIC-III datasets across different *ipc* levels for in-hospital mortality prediction. Best results at each *ipc* are highlighted in blue (ours) and red (baseline).

**(a) eICU dataset**

| Method | AUROC | | | | AUPRC | | | |
|---|---|---|---|---|---|---|---|---|
| | 50 | 100 | 200 | 500 | 50 | 100 | 200 | 500 |
| RANDOM | $0.740_{\pm0.022}$ | $0.763_{\pm0.013}$ | $0.804_{\pm0.008}$ | $0.840_{\pm0.005}$ | $0.277_{\pm0.027}$ | $0.308_{\pm0.022}$ | $0.368_{\pm0.017}$ | $0.429_{\pm0.013}$ |
| MTT | $0.754_{\pm0.003}$ | $0.800_{\pm0.001}$ | $0.829_{\pm0.004}$ | $0.849_{\pm0.001}$ | $0.329_{\pm0.003}$ | $0.390_{\pm0.004}$ | $0.398_{\pm0.007}$ | $0.454_{\pm0.002}$ |
| FTD | $0.798_{\pm0.002}$ | $0.803_{\pm0.002}$ | $0.824_{\pm0.001}$ | $0.847_{\pm0.004}$ | $0.374_{\pm0.002}$ | $0.389_{\pm0.001}$ | $0.430_{\pm0.001}$ | $0.442_{\pm0.004}$ |
| MCT | $0.773_{\pm0.004}$ | $0.828_{\pm0.001}$ | $0.837_{\pm0.002}$ | $0.847_{\pm0.001}$ | $0.397_{\pm0.005}$ | $0.427_{\pm0.001}$ | $0.448_{\pm0.001}$ | $0.464_{\pm0.001}$ |
| DATM | $0.854_{\pm0.001}$ | $0.852_{\pm0.001}$ | $0.846_{\pm0.001}$ | $0.856_{\pm0.001}$ | $0.462_{\pm0.001}$ | $0.451_{\pm0.003}$ | $0.464_{\pm0.001}$ | $0.482_{\pm0.001}$ |
| BTM (OURS) | $0.854_{\pm0.002}$ | $0.859_{\pm0.001}$ | $0.861_{\pm0.001}$ | $0.874_{\pm0.020}$ | $0.479_{\pm0.002}$ | $0.486_{\pm0.001}$ | $0.476_{\pm0.001}$ | $0.506_{\pm0.001}$ |
| **Full Dataset** | $0.879_{\pm0.002}$ | | | | $0.515_{\pm0.003}$ | | | |

**(b) MIMIC-III dataset**

| Method | AUROC | | | | AUPRC | | | |
|---|---|---|---|---|---|---|---|---|
| | 50 | 100 | 200 | 500 | 50 | 100 | 200 | 500 |
| RANDOM | $0.740_{\pm0.022}$ | $0.763_{\pm0.013}$ | $0.804_{\pm0.008}$ | $0.840_{\pm0.005}$ | $0.307_{\pm0.022}$ | $0.338_{\pm0.022}$ | $0.368_{\pm0.017}$ | $0.409_{\pm0.013}$ |
| MTT | $0.800_{\pm0.008}$ | $0.821_{\pm0.003}$ | $0.834_{\pm0.002}$ | $0.838_{\pm0.002}$ | $0.421_{\pm0.021}$ | $0.449_{\pm0.013}$ | $0.493_{\pm0.006}$ | $0.492_{\pm0.007}$ |
| FTD | $0.770_{\pm0.003}$ | $0.779_{\pm0.005}$ | $0.811_{\pm0.005}$ | $0.828_{\pm0.003}$ | $0.361_{\pm0.009}$ | $0.384_{\pm0.012}$ | $0.443_{\pm0.011}$ | $0.473_{\pm0.009}$ |
| MCT | $0.821_{\pm0.007}$ | $0.831_{\pm0.002}$ | $0.835_{\pm0.002}$ | $0.840_{\pm0.002}$ | $0.453_{\pm0.013}$ | $0.481_{\pm0.009}$ | $0.492_{\pm0.007}$ | $0.493_{\pm0.005}$ |
| DATM | $0.824_{\pm0.003}$ | $0.834_{\pm0.002}$ | $0.838_{\pm0.002}$ | $0.842_{\pm0.002}$ | $0.461_{\pm0.009}$ | $0.484_{\pm0.008}$ | $0.500_{\pm0.007}$ | $0.496_{\pm0.006}$ |
| BTM (OURS) | $0.828_{\pm0.005}$ | $0.835_{\pm0.003}$ | $0.839_{\pm0.002}$ | $0.840_{\pm0.002}$ | $0.471_{\pm0.008}$ | $0.488_{\pm0.007}$ | $0.499_{\pm0.006}$ | $0.503_{\pm0.003}$ |
| **Full Dataset** | $0.837_{\pm0.003}$ | | | | $0.499_{\pm0.008}$ | | | |

again suggests that structured low-rank surrogate supervision is particularly effective when the clinically relevant signal is sparse and must be retained under aggressive data compression.

On the eICU dataset, BTM attains the best or tied-best AUROC at every budget and the strongest AUPRC throughout. In AUPRC, the relative gains over the strongest baseline are 3.7%, 7.8%, 2.6%, and 5.0% at *ipc*=50, 100, 200, and 500, respectively. At the largest budget, BTM reaches 0.506, corresponding to 98.3% of full-dataset AUPRC (0.515). These results indicate that the learned Bézier surrogates retain task-relevant supervision particularly well even at low synthetic budgets.

On the MIMIC-III dataset, BTM achieves the best AUROC at *ipc*=50, 100, and 200, while remaining competitive at *ipc*=500. In AUPRC, BTM is best at *ipc*=50, 100, and 500, with relative gains of 2.2%, 0.8%, and 1.4% over the strongest baseline at those budgets. At *ipc*=200, DATM is marginally stronger by 0.001 AUPRC, indicating that both methods are highly competitive in this regime. At *ipc*=500, BTM reaches 0.503, slightly exceeding the full-dataset AUPRC of 0.499 within experimental variability.

Among the baselines, DATM and MCT are generally the strongest competitors on these public ICU benchmarks. However, BTM remains the most consistent overall in AUPRC, particularly on eICU, and is either best or near-best on MIMIC-III across all budgets. FTD appears less effective on MIMIC-III, which may reflect the structured time-series setting with a TCN, where encouraging uniformly flat trajectories can suppress temporally localised signals. Overall, the eICU and MIMIC-III results reinforce the same conclusion as the NHS cohorts that replacing raw SGD supervision with learned low-rank surrogate trajectories improves the utility of condensed datasets, especially in low-data settings.

**MIMIC-III phenotyping.** Table 4 reports results on the MIMIC-III dataset for 25-class multi-label phenotyping across synthetic budgets ranging from 50 to 500 instances per class, which we map to a per-label budget as described in Section 6. As in the binary prediction tasks, BTM performs strongly across all compression levels, with the clearest and most consistent gains appearing in macro AUPRC. This suggests that structured surrogate supervision is effective at preserving label-specific predictive signal under severe compression in the multi-label setting.

In terms of macro AUROC, BTM achieves the best performance at *ipc*=50, 100, and 200, with values of 0.690, 0.713, and 0.727, respectively. At *ipc*=500, DATM is marginally stronger (0.730 versus 0.729), indicating

Table 4: Performance on the MIMIC-III dataset for across different *ipc* levels 25-class multi-label phenotyping. Best results at each *ipc* are highlighted in blue (ours) and red (baseline).

**MIMIC-III dataset (Phenotyping)**

| Method | Macro AUROC | | | | Macro AUPRC | | | |
|---|---|---|---|---|---|---|---|---|
| | 50 | 100 | 200 | 500 | 50 | 100 | 200 | 500 |
| RANDOM | $0.602_{\pm 0.005}$ | $0.640_{\pm 0.005}$ | $0.661_{\pm 0.001}$ | $0.687_{\pm 0.002}$ | $0.261_{\pm 0.005}$ | $0.290_{\pm 0.004}$ | $0.312_{\pm 0.002}$ | $0.340_{\pm 0.002}$ |
| MTT | $0.668_{\pm 0.004}$ | $0.689_{\pm 0.004}$ | $0.705_{\pm 0.003}$ | $0.717_{\pm 0.003}$ | $0.310_{\pm 0.008}$ | $0.336_{\pm 0.006}$ | $0.358_{\pm 0.005}$ | $0.372_{\pm 0.004}$ |
| FTD | $0.659_{\pm 0.005}$ | $0.681_{\pm 0.004}$ | $0.699_{\pm 0.004}$ | $0.714_{\pm 0.003}$ | $0.299_{\pm 0.008}$ | $0.325_{\pm 0.007}$ | $0.341_{\pm 0.006}$ | $0.366_{\pm 0.005}$ |
| MCT | $0.675_{\pm 0.004}$ | $0.698_{\pm 0.004}$ | $0.713_{\pm 0.003}$ | $0.714_{\pm 0.002}$ | $0.321_{\pm 0.007}$ | $0.342_{\pm 0.006}$ | $0.361_{\pm 0.005}$ | $0.369_{\pm 0.004}$ |
| DATM | $0.679_{\pm 0.004}$ | $0.700_{\pm 0.003}$ | $0.715_{\pm 0.003}$ | $0.730_{\pm 0.002}$ | $0.327_{\pm 0.007}$ | $0.352_{\pm 0.006}$ | $0.371_{\pm 0.004}$ | $0.379_{\pm 0.003}$ |
| BTM (OURS) | $0.690_{\pm 0.003}$ | $0.713_{\pm 0.003}$ | $0.727_{\pm 0.002}$ | $0.729_{\pm 0.002}$ | $0.339_{\pm 0.006}$ | $0.365_{\pm 0.005}$ | $0.383_{\pm 0.004}$ | $0.389_{\pm 0.003}$ |
| **Full Dataset** | $0.738_{\pm 0.001}$ | | | | $0.395_{\pm 0.002}$ | | | |

that the two methods are highly competitive at larger synthetic budgets. Overall, the AUROC gains are smaller than in the binary tasks, which is consistent with the greater difficulty of matching supervision in the multi-label setting.

The improvements are more pronounced in macro AUPRC. BTM attains the best performance at every budget, exceeding the strongest baseline by 3.7%, 3.7%, 3.2%, and 2.7% at *ipc*=50, 100, 200, and 500, respectively. Macro AUPRC increases steadily from 0.339 at *ipc*=50 to 0.389 at *ipc*=500, reaching 98.5% of full-dataset performance (0.395). This pattern suggests that BTM is particularly effective at retaining weaker or less consistently reinforced label signals, which are especially important for precision–recall performance in multi-label clinical prediction.

Among the baselines, DATM is the strongest overall competitor. It remains consistently competitive across budgets and achieves the best macro AUROC at *ipc*=500. MCT is also competitive in macro AUROC, but its macro AUPRC is weaker, while MTT remains below DATM and BTM on both metrics. FTD is again less effective, particularly in macro AUPRC, suggesting that enforcing flatter surrogate trajectories alone is insufficient when supervision must preserve diverse, label-specific structure.

Overall, these results reinforce the same conclusion as in the binary tasks that replacing raw SGD supervision with structured, low-rank surrogate trajectories improves the utility of condensed datasets. In the multi-label setting, where supervision is inherently more diffuse because of label overlap, these benefits are most clearly reflected in macro AUPRC.

### 7.2 Cross-Architecture Generalisation

Cross-architecture evaluation tests whether the utility of a condensed dataset depends strongly on the architecture used during condensation. Figure 3 reports AUPRC at *ipc*=500. For each dataset, synthetic data are first condensed using a single source architecture, highlighted by the shaded region, and then evaluated on a set of unseen target architectures. We compare BTM only against DATM here because DATM was the strongest-performing baseline on these datasets in the main experiments, making it the most informative reference point for cross-architecture transfer.

On the eICU dataset, synthetic data condensed with an MLP transfer robustly across MLP variants of different depths and widths. BTM outperforms DATM on every target architecture, with absolute AUPRC gains ranging from 0.008 to 0.024. Its performance also remains stable across the MLP family, varying only from 0.488 to 0.506. This suggests that the supervision induced by BTM is not tightly coupled to a specific parametrisation of the MLP, but instead captures task-relevant signal that generalises across closely related architectures.

The transfer setting is more demanding on MIMIC-III, where condensation is performed with a TCN and evaluation includes both TCN and LSTM targets. Here, architecture shift changes the temporal inductive bias of the model, making transfer substantially harder. Even in this setting, BTM remains consistently stronger than DATM across all target architectures. The gains are modest within the TCN family, but larger under transfer to LSTMs, where BTM improves AUPRC by 0.012 and 0.028 on LSTM-1 and LSTM-2,

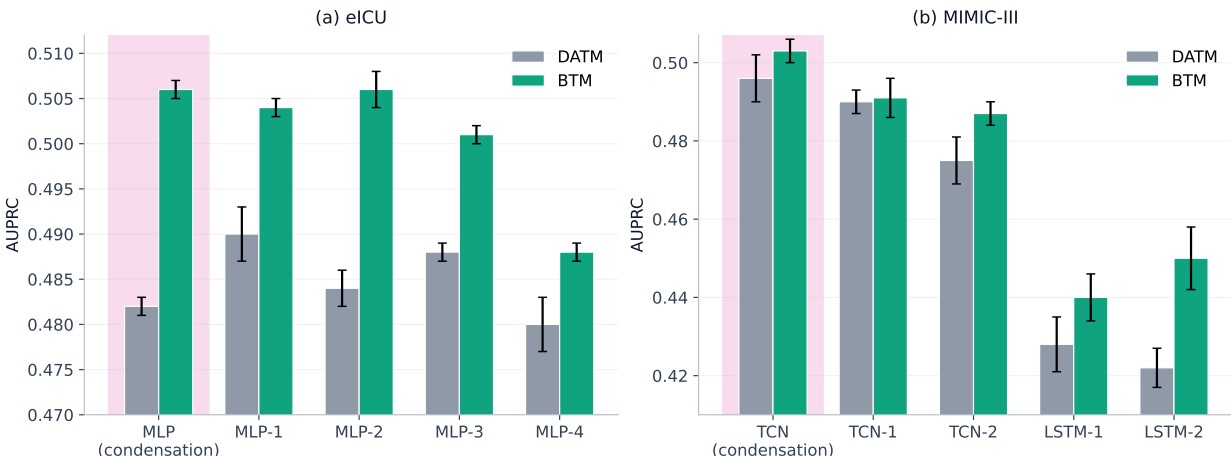

Figure 3: **Cross-architecture generalisation at $ipc=500$.** Synthetic datasets are condensed using a single source architecture for each dataset (shaded) and then evaluated on unseen target architectures. DATM is shown as the comparison baseline because it was the strongest-performing baseline on these datasets for in-hospital mortality prediction in the main experiments. BTM consistently outperforms DATM, especially under larger architecture shifts.

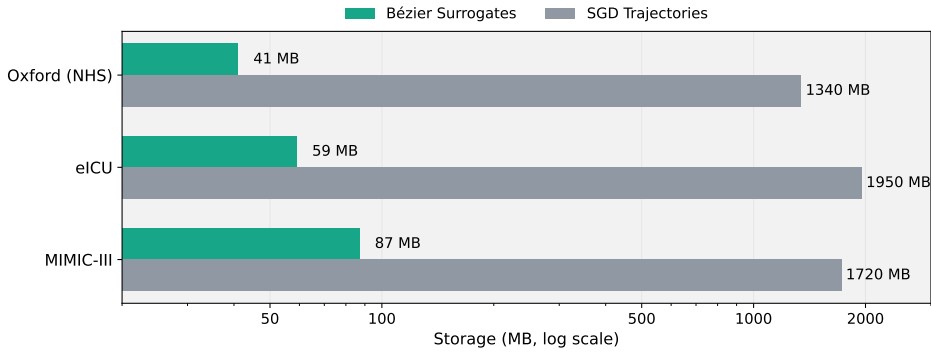

Figure 4: **Trajectory storage across clinical datasets.** Compared with full SGD trajectories, Bézier surrogates substantially reduce storage requirements, yielding approximately 33× lower storage on Oxford (NHS) and eICU, and 20× lower storage on MIMIC-III.

respectively. Relative to its source-architecture performance, BTM also exhibits slightly smaller degradation than DATM under TCN-to-LSTM transfer.

Overall, these results suggest that structuring the supervision signal through learned low-rank surrogate trajectories improves robustness to architecture shift. This effect is most evident when transferring across model classes with different inductive biases, where preserving task-relevant structure without overfitting to architecture-specific optimisation dynamics becomes especially important.

### 7.3 Storage Efficiency

Figure 4 shows that BTM substantially reduces trajectory storage across all clinical datasets. Relative to full SGD trajectories, Bézier surrogates yield approximately 33× lower storage on Oxford (NHS) and eICU, and 20× lower storage on MIMIC-III. These reductions lower memory requirements and make it easier to use more expert trajectories in resource-constrained clinical settings.

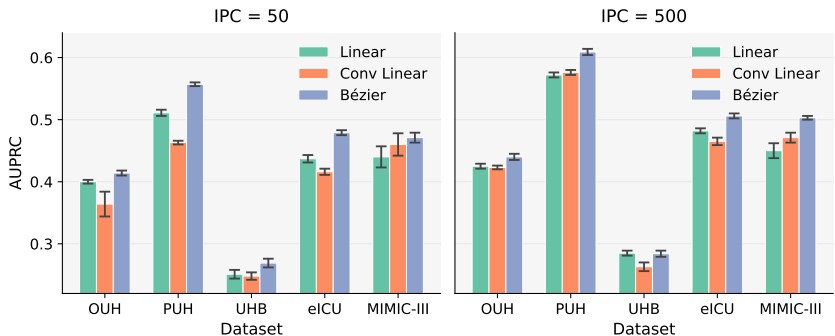

Figure 5: **Surrogate path complexity ablation.** AUPRC across five clinical datasets at $ipc$=50 and $ipc$=500 for three surrogate trajectory parameterisations: linear interpolation, convexified linear interpolation, and quadratic Bézier curves. OUH, PUH, and UHB denote Oxford, Portsmouth, and Birmingham NHS cohorts, respectively. Error bars denote standard deviation across runs. Bézier trajectories achieve the strongest overall performance, outperforming the linear variants in 9 of 10 dataset–budget settings.

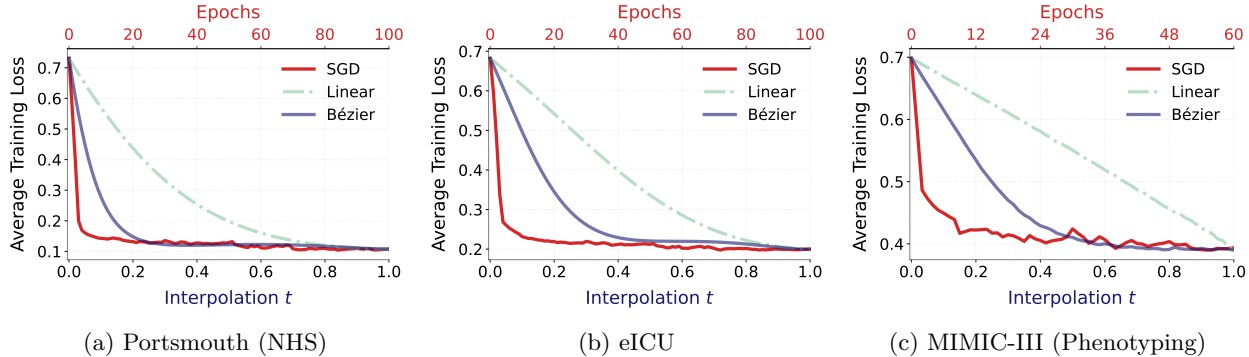

| (a) Portsmouth (NHS) | (b) eICU | (c) MIMIC-III (Phenotyping) |

Figure 6: **Training loss profiles along surrogate trajectories.** Average training loss for linear and quadratic Bézier trajectories as a function of interpolation parameter $t$, and for SGD as a function of training epochs, across datasets. While linear interpolation provides a smoother and more structured path than SGD, it can still traverse higher-loss regions. In contrast, the Bézier trajectory remains consistently in lower-loss regions due to its task-optimised construction.

## 7.4 Surrogate Path Complexity Ablation

Figure 5 evaluates how surrogate trajectory parameterisation affects condensation quality. We compare three increasingly structured path families: simple linear interpolation, a convexified linear trajectory that preserves temporal structure from SGD through fixed convex weights, and the proposed quadratic Bézier curve with a learnable control point. Results are reported in AUPRC across five clinical datasets at both low ($ipc$=50) and high ($ipc$=500) synthetic budgets.

A consistent trend emerges across datasets and budgets. Simple linear interpolation provides a strong baseline, indicating that reducing trajectory complexity is beneficial relative to storing and following full stochastic optimisation paths. Introducing a learnable control point yields further gains in almost every case: Bézier trajectories achieve the highest AUPRC in 9 of 10 dataset–budget settings and remain within 0.001 of the best result in the remaining case. Improvements are especially pronounced on the Oxford, Portsmouth, eICU, and MIMIC-III datasets, and persist across both low and high budgets.

By contrast, the convexified trajectory often underperforms simple linear interpolation, despite preserving more of the temporal structure of SGD. This suggests that retaining trajectory structure alone is insufficient; the path must be explicitly optimised to provide a compact and informative supervision signal.

Figure 6 provides a functional view of this effect by examining the loss landscape along each trajectory. While linear interpolation smooths the high-variance behaviour of SGD, it can still traverse higher-loss regions between endpoints. In contrast, the Bézier trajectory is optimised to minimise average loss along the path and remains consistently in lower-loss regions, yielding a supervision signal that is both structured and task-aligned rather than merely smoothed.

Together, these results support the central design choice of BTM: a low-complexity surrogate path is most effective when it is both structured and task-optimised, yielding a stronger and more informative supervision signal for dataset condensation.

## 8 Conclusion

We presented a geometric view of trajectory matching showing that, for a fixed synthetic dataset, the trajectory-matching residual is lower bounded by the component of the supervision signal lying outside the student's reachable gradient span. This yields a conditional rank-based representability bottleneck, where, when the supervision signal is not representable within that span, optimisation alone cannot eliminate the mismatch. This perspective clarifies a core limitation of existing trajectory-based dataset condensation methods and motivates the design of supervision signals that are better aligned with the student's optimisation geometry.

Motivated by this insight, we introduced Bézier Trajectory Matching (BTM), which replaces long stochastic optimisation trajectories with structured, low-rank surrogates. By learning quadratic Bézier paths that reduce average path loss and encourage a smooth descent profile from initialisation to solution, BTM provides a more structured supervision signal while preserving functionally meaningful training dynamics. Across diverse clinical datasets spanning tabular and time-series modalities, BTM consistently improves predictive performance, with the strongest gains in low-prevalence and low-data regimes. It also remains robust under architecture shift and reduces trajectory storage requirements by up to $33\times$, improving the practicality of trajectory-based condensation in resource-constrained clinical settings.

More broadly, our results suggest that effective dataset condensation depends not only on how much supervision is provided, but also on how that supervision is structured. In clinical contexts, this is particularly important because compact synthetic datasets may help support more efficient reuse of information derived from governed datasets while reducing storage and training costs. BTM currently uses a fixed quadratic parameterisation, and extending it to richer adaptive trajectory families is a natural next step. Incorporating differential privacy is another important direction for enabling safe and scalable clinical deployment.

**Broader Impact Statement**

This work has potential positive impact in machine learning settings where access to large real-world datasets is constrained by governance, storage, or computational cost. By improving dataset condensation, BTM may reduce the resources required to store, transfer, and repeatedly train on large datasets, thereby supporting more reproducible experimentation, lowering barriers for smaller research groups, and enabling more sustainable model development. In clinical machine learning, these benefits are especially relevant because access to governed health data is often restricted, so compact synthetic datasets may help support broader methodological research on information derived from such data in resource-constrained academic and clinical environments.

These benefits should be balanced against important risks. Condensed or synthetic datasets are not inherently safe to share, and BTM does not provide a formal privacy guarantee or replace privacy-preserving methods, data governance procedures, or access controls. Such datasets may also retain or amplify demographic imbalance, site-specific artefacts, and historical bias, which could lead to misleading conclusions or uneven downstream performance across patient subgroups. More capable condensation methods could also be misused to create compact surrogates of sensitive datasets that are redistributed or deployed without adequate scrutiny of privacy, fairness, or clinical validity. We therefore view BTM as a method for improving the utility of trajectory-based dataset condensation, rather than as a standalone solution for safe data sharing or clinical deployment. Any practical use should be accompanied by explicit privacy assessment, subgroup-level fairness

evaluation, and task-specific external validation. An important direction for future work is to combine structured trajectory-based condensation with formal privacy guarantees and stronger safeguards against bias, misuse, and over-interpretation of condensed data.

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

# A PIPELINE OF PROPOSED METHOD

---

**Algorithm 1** Dataset Condensation Using Trajectory Surrogates

---

**Require:** Collection of $M$ optimised Bézier surrogates $\{\Phi_{\boldsymbol{\phi}^\star}^{(m)}\}_{m=1}^M$, synthetic dataset $\tilde{\mathcal{D}} = \{(\tilde{\mathbf{x}}_i, \tilde{\mathbf{y}}_i)\}_{i=1}^{|\tilde{\mathcal{D}}|}$, number of classes $C$, input dimension $d$, student learning rate $\eta_s$, meta learning rate $\eta_{\mathbf{x}}$, number of student steps $N$, batch size $b$, maximum iterations $T_{\max}$

**Ensure:** Optimised synthetic dataset $\tilde{\mathcal{D}}^\star$
1: Stack synthetic inputs into matrix $\tilde{\mathbf{X}} \in \mathbb{R}^{|\tilde{\mathcal{D}}| \times d}$
2: Stack synthetic labels into tensor $\tilde{\mathbf{Y}} \in \mathbb{R}^{|\tilde{\mathcal{D}}| \times C}$
3: **for** $i = 1$ to $T_{\max}$ **do**
4:      Sample surrogate index $m \sim \mathcal{U}\{1, \ldots, M\}$
5:      Sample $t_s, t_e \sim \mathcal{U}(0,1)$ such that $t_s < t_e$
6:      $\boldsymbol{\theta}_s \leftarrow \Phi_{\boldsymbol{\phi}^\star}^{(m)}(t_s)$ {Start position on surrogate}
7:      $\boldsymbol{\theta}_e \leftarrow \Phi_{\boldsymbol{\phi}^\star}^{(m)}(t_e)$ {Target position on surrogate}
8:      $\hat{\boldsymbol{\theta}}_0 \leftarrow \boldsymbol{\theta}_s$ {Initialise student model}
9:      **for** $n = 0$ to $N - 1$ **do**
10:          Sample mini-batch $B_n \subset \tilde{\mathcal{D}}$, $|B_n| = b$
11:          $\hat{\boldsymbol{\theta}}_{n+1} \leftarrow \hat{\boldsymbol{\theta}}_n - \eta_s \nabla_{\hat{\boldsymbol{\theta}}_n} \left[ \frac{1}{b} \sum_{\substack{(\tilde{\mathbf{x}}, \tilde{\mathbf{y}}) \\ \in B_n}} \ell(\hat{\boldsymbol{\theta}}_n; \tilde{\mathbf{x}}, \tilde{\mathbf{y}}) \right]$
12:      **end for**
13:      $\mathcal{L}_{\mathrm{BTM}} \leftarrow \dfrac{\|\hat{\boldsymbol{\theta}}_N - \boldsymbol{\theta}_e\|_2^2}{\|\boldsymbol{\theta}_s - \boldsymbol{\theta}_e\|_2^2}$ {Normalised matching loss}
14:      $\boldsymbol{g}_L \leftarrow \nabla_{\hat{\boldsymbol{\theta}}_N} \mathcal{L}_{\mathrm{BTM}} = \dfrac{2(\hat{\boldsymbol{\theta}}_N - \boldsymbol{\theta}_e)}{\|\boldsymbol{\theta}_s - \boldsymbol{\theta}_e\|_2^2}$ {Gradient signal}
15:      Compute $\nabla_{\tilde{\mathbf{X}}} \mathcal{L}_{\mathrm{BTM}} \approx -\eta_s \sum_{n=0}^{N-1} \frac{1}{|B_n|} \sum_{(\tilde{\mathbf{x}}, \tilde{\mathbf{y}}) \in B_n} \nabla_{\tilde{\mathbf{X}}} \left\langle \nabla_{\boldsymbol{\theta}} \ell(\hat{\boldsymbol{\theta}}_n; \tilde{\mathbf{x}}, \tilde{\mathbf{y}}), \boldsymbol{g}_L \right\rangle$,
16:      $\tilde{\mathbf{X}} \leftarrow \tilde{\mathbf{X}} - \eta_{\mathbf{x}} \nabla_{\tilde{\mathbf{X}}} \mathcal{L}_{\mathrm{BTM}}$ {Update synthetic inputs}
17: **end for**
18: **return** $\tilde{\mathcal{D}}^\star = \tilde{\mathcal{D}}$

---

# B Proofs

This section gathers the proofs of the main theoretical results. We first prove the geometric lower bounds underlying trajectory matching, namely Theorem 1 and Theorem 2, which formalise the restriction imposed by the student's reachable span. We then prove Theorem 3, which establishes the low-dimensional structure of the displacements induced by quadratic Bézier surrogate trajectories. Finally, we prove Theorem 4 and Corollary 1. The corollary is a conditional statement rather than an algorithmic guarantee; it isolates a sufficient condition under which a synthetic trajectory would inherit the surrogate's prediction-space behaviour.

Throughout, we retain the notation introduced in the main text. All norms on parameter vectors are Euclidean unless stated otherwise, and $\|\cdot\|_F$ denotes the Frobenius norm for matrices.

## B.1 Proof of Theorem 1

*Proof.* Fix a segment $k$. Under the first-order, no-momentum student update

$$\hat{\boldsymbol{\theta}}_{n+1} = \hat{\boldsymbol{\theta}}_n - \eta_s \mathbf{g}_n, \qquad n = 0, \ldots, N-1, \tag{1}$$

the student displacement after $N$ inner steps is

$$\boldsymbol{\delta}_k = \hat{\boldsymbol{\theta}}_N - \boldsymbol{\theta}_{s_k} = -\eta_s \sum_{n=0}^{N-1} \mathbf{g}_n. \tag{2}$$

Since each $\mathbf{g}_n$ belongs to the reachable gradient span

$$\mathcal{G}_k = \mathrm{span}\{\mathbf{g}_0, \ldots, \mathbf{g}_{N-1}\}, \tag{3}$$

it follows immediately that

$$\boldsymbol{\delta}_k \in \mathcal{G}_k. \tag{4}$$

By definition, the trajectory matching loss for segment $k$ is

$$\mathcal{L}_{\mathrm{TM}}(s_k, e_k; \tilde{\mathcal{D}}) = \|\boldsymbol{\delta}_k - \boldsymbol{\Delta}_k\|_2^2. \tag{5}$$

Thus, for fixed teacher displacement $\boldsymbol{\Delta}_k$, the loss is the squared distance from $\boldsymbol{\Delta}_k$ to a point $\boldsymbol{\delta}_k$ constrained to lie in $\mathcal{G}_k$. Therefore

$$\mathcal{L}_{\mathrm{TM}}(s_k, e_k; \tilde{\mathcal{D}}) \geq \min_{\mathbf{v} \in \mathcal{G}_k} \|\mathbf{v} - \boldsymbol{\Delta}_k\|_2^2. \tag{6}$$

Let $P_{\mathcal{G}_k}$ denote the orthogonal projector onto $\mathcal{G}_k$. The orthogonal decomposition of $\boldsymbol{\Delta}_k$ with respect to $\mathcal{G}_k$ is

$$\boldsymbol{\Delta}_k = P_{\mathcal{G}_k}\boldsymbol{\Delta}_k + (I - P_{\mathcal{G}_k})\boldsymbol{\Delta}_k, \tag{7}$$

where

$$P_{\mathcal{G}_k}\boldsymbol{\Delta}_k \in \mathcal{G}_k, \qquad (I - P_{\mathcal{G}_k})\boldsymbol{\Delta}_k \in \mathcal{G}_k^{\perp}. \tag{8}$$

Hence, for any $\mathbf{v} \in \mathcal{G}_k$,

$$\mathbf{v} - \boldsymbol{\Delta}_k = \big(\mathbf{v} - P_{\mathcal{G}_k}\boldsymbol{\Delta}_k\big) - (I - P_{\mathcal{G}_k})\boldsymbol{\Delta}_k. \tag{9}$$

The first term lies in $\mathcal{G}_k$, whereas the second lies in $\mathcal{G}_k^{\perp}$, so the two terms are orthogonal. By the Pythagorean theorem,

$$\|\mathbf{v} - \boldsymbol{\Delta}_k\|_2^2 = \|\mathbf{v} - P_{\mathcal{G}_k}\boldsymbol{\Delta}_k\|_2^2 + \|(I - P_{\mathcal{G}_k})\boldsymbol{\Delta}_k\|_2^2 \geq \|(I - P_{\mathcal{G}_k})\boldsymbol{\Delta}_k\|_2^2. \tag{10}$$

Taking $\mathbf{v} = \boldsymbol{\delta}_k \in \mathcal{G}_k$ yields

$$\mathcal{L}_{\mathrm{TM}}(s_k, e_k; \tilde{\mathcal{D}}) = \|\boldsymbol{\delta}_k - \boldsymbol{\Delta}_k\|_2^2 \geq \|(I - P_{\mathcal{G}_k})\boldsymbol{\Delta}_k\|_2^2. \tag{11}$$

It remains to identify the minimiser of the best-in-subspace problem. Equality in

$$\|\mathbf{v} - \boldsymbol{\Delta}_k\|_2^2 = \|\mathbf{v} - P_{\mathcal{G}_k}\boldsymbol{\Delta}_k\|_2^2 + \|(I - P_{\mathcal{G}_k})\boldsymbol{\Delta}_k\|_2^2 \tag{12}$$

holds if and only if

$$\|\mathbf{v} - P_{\mathcal{G}_k}\boldsymbol{\Delta}_k\|_2^2 = 0, \tag{13}$$

that is, if and only if

$$\mathbf{v} = P_{\mathcal{G}_k}\boldsymbol{\Delta}_k. \tag{14}$$

Therefore the minimiser of $\min_{\mathbf{v} \in \mathcal{G}_k} \|\mathbf{v} - \boldsymbol{\Delta}_k\|_2^2$ is

$$\mathbf{v}^{\star} = P_{\mathcal{G}_k}\boldsymbol{\Delta}_k. \tag{15}$$

This identifies the tightest point in the reachable span. Whether the realised student displacement $\boldsymbol{\delta}_k$ attains this minimiser depends on the synthetic-data optimisation and is not asserted here. This proves the theorem. $\square$

### B.2  Proof of Theorem 2

*Proof.* For each segment $k$, the student displacement satisfies

$$\boldsymbol{\delta}_k \in \mathcal{G}_k. \tag{16}$$

Define the global reachable span by

$$\mathcal{G} = \mathrm{span}\{\mathbf{g}_n^{(k)} : k = 1, \ldots, K, \ n = 0, \ldots, N-1\}. \tag{17}$$

Since

$$\mathcal{G}_k = \mathrm{span}\{\mathbf{g}_0^{(k)}, \ldots, \mathbf{g}_{N-1}^{(k)}\},\tag{18}$$

it follows that $\mathcal{G}_k \subseteq \mathcal{G}$ for every $k$. Hence

$$\boldsymbol{\delta}_k \in \mathcal{G} \qquad \text{for all } k = 1, \ldots, K.\tag{19}$$

Define

$$\mathcal{L}_{\mathrm{TM}}^{(k)} := \mathcal{L}_{\mathrm{TM}}(s_k, e_k; \tilde{\mathcal{D}}) = \|\boldsymbol{\delta}_k - \boldsymbol{\Delta}_k\|_2^2.\tag{20}$$

Since $\boldsymbol{\delta}_k \in \mathcal{G}$, this loss is the squared distance from $\boldsymbol{\Delta}_k$ to a point in $\mathcal{G}$. Exactly as in the proof of Theorem 1, if $P_{\mathcal{G}}$ denotes the orthogonal projector onto $\mathcal{G}$, then

$$\mathcal{L}_{\mathrm{TM}}^{(k)} \geq \min_{\mathbf{v} \in \mathcal{G}} \|\mathbf{v} - \boldsymbol{\Delta}_k\|_2^2 = \|(I - P_{\mathcal{G}})\boldsymbol{\Delta}_k\|_2^2.\tag{21}$$

Thus, for every $k$,

$$\mathcal{L}_{\mathrm{TM}}^{(k)} \geq \|(I - P_{\mathcal{G}})\boldsymbol{\Delta}_k\|_2^2.\tag{22}$$

Summing over $k = 1, \ldots, K$ gives

$$\sum_{k=1}^{K} \mathcal{L}_{\mathrm{TM}}^{(k)} \geq \sum_{k=1}^{K} \|(I - P_{\mathcal{G}})\boldsymbol{\Delta}_k\|_2^2.\tag{23}$$

Now recall the teacher displacement matrix

$$\mathbf{A} = [\boldsymbol{\Delta}_1, \ldots, \boldsymbol{\Delta}_K] \in \mathbb{R}^{p \times K}.\tag{24}$$

Then

$$(I - P_{\mathcal{G}})\mathbf{A} = \big[(I - P_{\mathcal{G}})\boldsymbol{\Delta}_1, \ldots, (I - P_{\mathcal{G}})\boldsymbol{\Delta}_K\big].\tag{25}$$

Using the identity

$$\|\mathbf{M}\|_F^2 = \sum_{k=1}^{K} \|\mathbf{M}_{:,k}\|_2^2\tag{26}$$

for any matrix $\mathbf{M} \in \mathbb{R}^{p \times K}$, we obtain

$$\sum_{k=1}^{K} \|(I - P_{\mathcal{G}})\boldsymbol{\Delta}_k\|_2^2 = \|(I - P_{\mathcal{G}})\mathbf{A}\|_F^2.\tag{27}$$

Therefore

$$\sum_{k=1}^{K} \mathcal{L}_{\mathrm{TM}}^{(k)} \geq \|(I - P_{\mathcal{G}})\mathbf{A}\|_F^2,\tag{28}$$

and dividing by $K$ yields

$$\frac{1}{K} \sum_{k=1}^{K} \mathcal{L}_{\mathrm{TM}}^{(k)} \geq \frac{1}{K} \|(I - P_{\mathcal{G}})\mathbf{A}\|_F^2.\tag{29}$$

It remains to minimise this residual over all $r$-dimensional subspaces $\mathcal{G} \subset \mathbb{R}^p$. Let the singular value decomposition of $\mathbf{A}$ be

$$\mathbf{A} = \mathbf{U}\boldsymbol{\Sigma}\mathbf{V}^\top,\tag{30}$$

with singular values ordered as

$$\sigma_1(\mathbf{A}) \geq \sigma_2(\mathbf{A}) \geq \cdots \geq \sigma_{r_\mathbf{A}}(\mathbf{A}) > 0,\tag{31}$$

where

$$r_\mathbf{A} := \mathrm{rank}(\mathbf{A}).\tag{32}$$

The optimisation problem

$$\min_{\dim(\mathcal{G})=r} \|(I - P_{\mathcal{G}})\mathbf{A}\|_F^2 \tag{33}$$

is equivalent to the problem of finding the best rank-$r$ approximation to $\mathbf{A}$ in Frobenius norm, since $P_{\mathcal{G}}\mathbf{A}$ has all of its columns in $\mathcal{G}$ and therefore has rank at most $r$. By the Eckart–Young–Mirsky theorem, the optimal choice is to take $\mathcal{G}$ to be the span of the top $r$ left singular vectors of $\mathbf{A}$, in which case the minimum residual is

$$\min_{\dim(\mathcal{G})=r} \|(I - P_{\mathcal{G}})\mathbf{A}\|_F^2 = \sum_{j=r+1}^{r_{\mathbf{A}}} \sigma_j(\mathbf{A})^2. \tag{34}$$

Dividing by $K$ gives

$$\min_{\dim(\mathcal{G})=r} \frac{1}{K}\|(I - P_{\mathcal{G}})\mathbf{A}\|_F^2 = \frac{1}{K}\sum_{j=r+1}^{r_{\mathbf{A}}} \sigma_j(\mathbf{A})^2. \tag{35}$$

This proves the theorem. $\qquad\square$

### B.3   Proof of Theorem 3

*Proof.* Consider the quadratic Bézier trajectory

$$\Phi(t) = (1-t)^2 \boldsymbol{\theta}_0 + 2t(1-t)\boldsymbol{\phi} + t^2 \boldsymbol{\theta}_T, \qquad t \in [0,1]. \tag{36}$$

For any $0 \le t_s < t_e \le 1$, define the segment displacement

$$\boldsymbol{\Delta}(t_s, t_e) := \Phi(t_e) - \Phi(t_s). \tag{37}$$

We begin by rewriting $\Phi(t)$ in a form that makes its underlying linear structure explicit. Expanding the quadratic terms gives

$$\Phi(t) = \boldsymbol{\theta}_0 + 2t(\boldsymbol{\phi} - \boldsymbol{\theta}_0) + t^2(\boldsymbol{\theta}_0 - 2\boldsymbol{\phi} + \boldsymbol{\theta}_T). \tag{38}$$

Hence, for any $t_s, t_e \in [0,1]$,

$$\begin{aligned}
\boldsymbol{\Delta}(t_s, t_e) &= 2(t_e - t_s)(\boldsymbol{\phi} - \boldsymbol{\theta}_0) + (t_e^2 - t_s^2)(\boldsymbol{\theta}_0 - 2\boldsymbol{\phi} + \boldsymbol{\theta}_T) \\
&= (t_e - t_s)\Big[2(\boldsymbol{\phi} - \boldsymbol{\theta}_0) + (t_e + t_s)(\boldsymbol{\theta}_0 - 2\boldsymbol{\phi} + \boldsymbol{\theta}_T)\Big].
\end{aligned} \tag{39}$$

Therefore every segment displacement belongs to the subspace

$$\mathrm{span}\big\{\boldsymbol{\phi} - \boldsymbol{\theta}_0,\ \boldsymbol{\theta}_0 - 2\boldsymbol{\phi} + \boldsymbol{\theta}_T\big\}. \tag{40}$$

By definition,

$$V_\Phi := \mathrm{span}\big\{\Phi(t_e) - \Phi(t_s):\ 0 \le t_s < t_e \le 1\big\}, \tag{41}$$

so we conclude that

$$V_\Phi \subseteq \mathrm{span}\big\{\boldsymbol{\phi} - \boldsymbol{\theta}_0,\ \boldsymbol{\theta}_0 - 2\boldsymbol{\phi} + \boldsymbol{\theta}_T\big\}. \tag{42}$$

Since the latter space has dimension at most 2, it follows that

$$\dim(V_\Phi) \le 2. \tag{43}$$

This proves the first claim. For the second claim, let

$$\mathbf{A}_\Phi = [\boldsymbol{\Delta}(t_s^{(1)}, t_e^{(1)}), \ldots, \boldsymbol{\Delta}(t_s^{(K)}, t_e^{(K)})] \in \mathbb{R}^{p \times K} \tag{44}$$

be any displacement matrix formed from segments of $\Phi$. Each column of $\mathbf{A}_\Phi$ lies in $V_\Phi$, and we have shown that $\dim(V_\Phi) \le 2$. Hence the column space of $\mathbf{A}_\Phi$ has dimension at most 2, so

$$\mathrm{rank}(\mathbf{A}_\Phi) \le 2. \tag{45}$$

Therefore any displacement matrix constructed from segments of a single quadratic Bézier trajectory has rank at most 2. $\qquad\square$

### B.4 Auxiliary lemma for Theorem 4

We first characterise the deviation of a quadratic Bézier curve from the chord joining its endpoints.

**Lemma 1.** *Let*

$$c(t) := (1 - t)\boldsymbol{\theta}_0 + t\boldsymbol{\theta}_T. \tag{46}$$

*Then, for all $t \in [0, 1]$,*

$$\Phi(t) - c(t) = t(1 - t)\big(2\boldsymbol{\phi}^\star - \boldsymbol{\theta}_0 - \boldsymbol{\theta}_T\big), \tag{47}$$

*and hence*

$$\|\Phi(t) - c(t)\|_2 = \frac{t(1 - t)}{2}\kappa. \tag{48}$$

*In particular,*

$$\sup_{t \in [0,1]} \|\Phi(t) - c(t)\|_2 = \frac{\kappa}{8}. \tag{49}$$

*Proof.* Using the definitions of $\Phi(t)$ and $c(t)$,

$$\Phi(t) - c(t) = \Big((1 - t)^2 - (1 - t)\Big)\boldsymbol{\theta}_0 + 2t(1 - t)\boldsymbol{\phi}^\star + \Big(t^2 - t\Big)\boldsymbol{\theta}_T. \tag{50}$$

Since

$$(1 - t)^2 - (1 - t) = -t(1 - t), \qquad t^2 - t = -t(1 - t), \tag{51}$$

we obtain

$$\Phi(t) - c(t) = t(1 - t)\big(2\boldsymbol{\phi}^\star - \boldsymbol{\theta}_0 - \boldsymbol{\theta}_T\big). \tag{52}$$

Taking Euclidean norms and using the definition

$$\kappa = 2\|\boldsymbol{\theta}_0 - 2\boldsymbol{\phi}^\star + \boldsymbol{\theta}_T\|_2 = 2\|2\boldsymbol{\phi}^\star - \boldsymbol{\theta}_0 - \boldsymbol{\theta}_T\|_2, \tag{53}$$

gives

$$\|\Phi(t) - c(t)\|_2 = \frac{t(1 - t)}{2}\kappa. \tag{54}$$

Finally, $t(1 - t) \leq \frac{1}{4}$ for all $t \in [0, 1]$, with equality at $t = \frac{1}{2}$. Therefore

$$\sup_{t \in [0,1]} \|\Phi(t) - c(t)\|_2 = \frac{\kappa}{8}. \tag{55}$$

$\square$

### B.5 Proof of Theorem 4

*Proof.* We prove the two claims separately.

**(i) Smooth curvature.** Consider the optimised quadratic Bézier surrogate

$$\Phi(t) = (1 - t)^2\boldsymbol{\theta}_0 + 2t(1 - t)\boldsymbol{\phi}^\star + t^2\boldsymbol{\theta}_T. \tag{56}$$

Differentiating twice with respect to $t$ yields

$$\Phi''(t) = 2(\boldsymbol{\theta}_0 - 2\boldsymbol{\phi}^\star + \boldsymbol{\theta}_T), \tag{57}$$

which is constant in $t$. Hence

$$\sup_{t \in [0,1]} \|\Phi''(t)\|_2 = 2\|\boldsymbol{\theta}_0 - 2\boldsymbol{\phi}^\star + \boldsymbol{\theta}_T\|_2 = \kappa. \tag{58}$$

The chord-deviation bound in Theorem 4(i) follows from Lemma 1.

**(ii) Prediction fidelity.** Fix any $\boldsymbol{x} \in \mathcal{X}$ and $t \in [0, 1]$. By the $L_f$-Lipschitz continuity of the model map in parameter space on the set traced by $\Phi$ and $\gamma$,

$$\|f_{\Phi(t)}(\boldsymbol{x}) - f_{\gamma(t)}(\boldsymbol{x})\|_2 \le L_f \|\Phi(t) - \gamma(t)\|_2. \tag{59}$$

We now bound the parameter-space discrepancy by introducing the endpoint chord $c(t)$. By the triangle inequality,

$$\|\Phi(t) - \gamma(t)\|_2 \le \|\Phi(t) - c(t)\|_2 + \|\gamma(t) - c(t)\|_2. \tag{60}$$

By Lemma 1,

$$\|\Phi(t) - c(t)\|_2 \le \frac{\kappa}{8}, \tag{61}$$

and by definition of $D$,

$$\|\gamma(t) - c(t)\|_2 \le D. \tag{62}$$

Therefore

$$\|\Phi(t) - \gamma(t)\|_2 \le \frac{\kappa}{8} + D, \tag{63}$$

and consequently

$$\|f_{\Phi(t)}(\boldsymbol{x}) - f_{\gamma(t)}(\boldsymbol{x})\|_2 \le L_f \left( \frac{\kappa}{8} + D \right). \tag{64}$$

Taking the supremum over $\boldsymbol{x} \in \mathcal{X}$ and $t \in [0, 1]$ proves the claim. $\qquad\square$

### B.5.1 Teacher-path curvature intuition.

For a generic teacher update sequence of the form

$$\boldsymbol{\theta}_{\tau+1} = \boldsymbol{\theta}_\tau - \eta \mathbf{u}_\tau, \tag{65}$$

define the discrete second-order difference

$$\mathbf{h}_\tau := \boldsymbol{\theta}_{\tau+1} - 2\boldsymbol{\theta}_\tau + \boldsymbol{\theta}_{\tau-1}. \tag{66}$$

Substituting the update at times $\tau$ and $\tau - 1$ gives

$$\mathbf{h}_\tau = -\eta(\mathbf{u}_\tau - \mathbf{u}_{\tau-1}). \tag{67}$$

If $\mathbf{u}_\tau = \mathbf{d}_\tau + \boldsymbol{\xi}_\tau$ is decomposed into systematic drift and stochastic variability, then

$$\mathbf{h}_\tau = -\eta \big[ (\mathbf{d}_\tau - \mathbf{d}_{\tau-1}) + (\boldsymbol{\xi}_\tau - \boldsymbol{\xi}_{\tau-1}) \big]. \tag{68}$$

Thus, discrete teacher curvature mixes optimisation drift with stochastic variability, whereas the quadratic Bézier surrogate has globally smooth curvature.

### B.6 Proof of Corollary 1

*Proof.* Fix any $\boldsymbol{x} \in \mathcal{X}$ and $t \in [0, 1]$. By the triangle inequality,

$$\|f_{\gamma_{\mathrm{stu}}(t)}(\boldsymbol{x}) - f_{\gamma(t)}(\boldsymbol{x})\|_2 \le \|f_{\gamma_{\mathrm{stu}}(t)}(\boldsymbol{x}) - f_{\Phi(t)}(\boldsymbol{x})\|_2 + \|f_{\Phi(t)}(\boldsymbol{x}) - f_{\gamma(t)}(\boldsymbol{x})\|_2. \tag{69}$$

Using again the $L_f$-Lipschitz continuity of the model map,

$$\|f_{\gamma_{\mathrm{stu}}(t)}(\boldsymbol{x}) - f_{\Phi(t)}(\boldsymbol{x})\|_2 \le L_f \|\gamma_{\mathrm{stu}}(t) - \Phi(t)\|_2 \le L_f \varepsilon_{\mathrm{syn}}. \tag{70}$$

Moreover, Theorem 4(ii) gives

$$\|f_{\Phi(t)}(\boldsymbol{x}) - f_{\gamma(t)}(\boldsymbol{x})\|_2 \le L_f \left( \frac{\kappa}{8} + D \right). \tag{71}$$

Combining the two bounds yields

$$\|f_{\gamma_{\mathrm{stu}}(t)}(\boldsymbol{x}) - f_{\gamma(t)}(\boldsymbol{x})\|_2 \le L_f \left( \varepsilon_{\mathrm{syn}} + \frac{\kappa}{8} + D \right). \tag{72}$$

Taking the supremum over $\boldsymbol{x} \in \mathcal{X}$ and $t \in [0, 1]$ proves the corollary. $\qquad\square$

# C  DATASET DETAILS

We provide implementation-specific preprocessing and representation details for each dataset. Shared experimental settings (splits, normalisation, and evaluation protocol) are described in Section 6.

## C.1  NHS Emergency Department Datasets (Oxford, Portsmouth, Birmingham)

For the NHS cohorts, each patient is represented by a fixed 27-dimensional feature vector derived from admission-time data. Missing values are imputed using feature-wise medians computed on the training set.

All features correspond to measurements available at presentation, and no temporal aggregation is performed. Each dataset is treated independently, with no cross-site data mixing.

Dataset-level statistics are summarised in Table 1. Each dataset comprises routinely collected admission-time features, including demographics, vital signs, and laboratory blood tests, resulting in a total of 27 features per patient. The clinical predictors are grouped by category in Table 2. Missing values are handled via median imputation.

Access to the datasets is governed by NHS and Health Research Authority (HRA) approval (IRAS ID: 281832). Oxford data are available via the Infections in Oxfordshire Research Database[1], while Portsmouth and Birmingham data are accessible upon reasonable request to the respective trusts.

Table 1: Dataset characteristics for NHS emergency department datasets.

|  | Oxford | Portsmouth | Birmingham |
|---|---|---|---|
| # Examples | 161,955 | 38,717 | 95,236 |
| # Positive Examples | 2,791 | 2,005 | 790 |
| # Features | 27 | 27 | 27 |
| Prevalence (%) | 1.7 | 5.3 | 0.8 |

Table 2: Clinical predictors used for COVID-19 diagnosis.

| Category | Features |
|---|---|
| Vital Signs | Heart rate, respiratory rate, oxygen saturation, systolic blood pressure, diastolic blood pressure, temperature |
| Blood Tests | Haemoglobin, haematocrit, mean cell volume, white cell count, neutrophil count, lymphocyte count, monocyte count, eosinophil count, basophil count, platelets |
| Liver Function Tests & C-reactive protein | Albumin, alkaline phosphatase, alanine aminotransferase, bilirubin, C-reactive protein |
| Urea & Electrolytes | Sodium, potassium, creatinine, urea, estimated glomerular filtration rate |

## C.2  eICU

We use the pre-processed version of the eICU Collaborative Research Database available at https://physionet.org/content/mimic-eicu-fiddle-feature/1.0.0/. This representation maps each ICU stay to a fixed 402-dimensional feature vector.

Dataset characteristics are reported in Table 3 and variables used to construct the feature representation are summarised in Table 4.

---

[1] https://oxfordbrc.nihr.ac.uk/research-themes/modernising-medical-microbiology-and-big-infection-diagnostics/infections-in-oxfordshire-research-database-iord/

Continuous variables are discretised into quantile-based bins (typically five bins), and summary statistics (minimum, maximum, and mean) are computed where applicable. Categorical variables are one-hot encoded. This results in a fully tabular representation with no temporal dimension.

Table 3: Dataset characteristics for eICU.

|  | eICU |
|---|---|
| # Examples | 49,305 |
| # Positive Examples | 4,501 |
| # Features | 402 |
| Prevalence (%) | 9.1 |

Table 4: Source physiological and demographic variables used to construct the 402-dimensional feature vector for eICU in-hospital mortality prediction.

| Category | Variables |
|---|---|
| Demographics | Age, height, weight, gender |
| Admission Information | Hospital admit source, hospital admit offset, unit admit source, unit stay type, unit type, Apache admission diagnosis, airway type |
| Vital Signs | Heart rate, respiratory rate, oxygen saturation, temperature (Celsius and Fahrenheit), temperature location |
| Blood Pressure | Non-invasive systolic/diastolic/mean blood pressure, invasive systolic/diastolic/mean blood pressure, central venous pressure |
| Oxygen Support | O2 administration device, O2 level percentage |
| Laboratory Measurements | Glucose |

### C.3 MIMIC-III

We process MIMIC-III using publicly available benchmarking code to obtain multivariate time-series representations. Only patients with at least 48 hourly observations are retained, and the first 48 hours of each stay are used for all tasks.

After removing duplicate features, each time step is represented by a 60-dimensional feature vector, yielding an input tensor of shape $48 \times 60$ per patient. Binary mask features are included to indicate the presence or absence of each measurement at each time step.

Categorical clinical variables are encoded as follows: Glasgow Coma Scale components are one-hot encoded (eye opening: 5 categories; motor response: 6 categories; verbal response: 5 categories; total score: 11 categories), and capillary refill rate is encoded as a binary variable.

The in-hospital mortality (IHM) prediction and phenotyping tasks follow the standard benchmark settings, with data characteristics summarised in 5. For completeness, we list below the input variables used in MIMIC-III as well as the 25 phenotyping classes.

Table 5: Dataset characteristics for MIMIC-III.

|  | MIMIC-III (IHM) | MIMIC-III (Phenotyping) |
|---|---|---|
| # Examples | 21,156 | 21,728 |
| # Time-steps | 48 | 48 |
| # Features per time-step | 60 | 60 |

**List of features in MIMIC-III dataset**

1. Capillary refill rate-0.0
2. Capillary refill rate-1.0
3. Diastolic blood pressure
4. Fraction inspired oxygen
5. Glascow coma scale eye opening-2 To Pain
6. Glascow coma scale eye opening-3 To speech
7. Glascow coma scale eye opening-1 No Response
8. Glascow coma scale eye opening-4 Spontaneously
9. Glascow coma scale eye opening-0 None
10. Glascow coma scale motor response-1 No Movement
11. Glascow coma scale motor response-3 Abnormal flexion
12. Glascow coma scale motor response-2 Abnormal extension
13. Glascow coma scale motor response-4 Flex-withdraws
14. Glascow coma scale motor response-5 Localizes Pain
15. Glascow coma scale motor response-6 Obeys Commands
16. Glascow coma scale total-11
17. Glascow coma scale total-10
18. Glascow coma scale total-13
19. Glascow coma scale total-12
20. Glascow coma scale total-15
21. Glascow coma scale total-14
22. Glascow coma scale total-3
23. Glascow coma scale total-5
24. Glascow coma scale total-4
25. Glascow coma scale total-7
26. Glascow coma scale total-6
27. Glascow coma scale total-9
28. Glascow coma scale total-8
29. Glascow coma scale verbal response-1 No Response
30. Glascow coma scale verbal response-4 Confused
31. Glascow coma scale verbal response-2 Incomprehensible sounds
32. Glascow coma scale verbal response-3 Inappropriate Words
33. Glascow coma scale verbal response-5 Oriented
34. Glucose
35. Heart Rate
36. Height
37. Mean blood pressure
38. Oxygen saturation
39. Respiratory rate
40. Systolic blood pressure
41. Temperature
42. Weight
43. pH
44. mask-Capillary refill rate
45. mask-Diastolic blood pressure
46. mask-Fraction inspired oxygen
47. mask-Glascow coma scale eye opening
48. mask-Glascow coma scale motor response
49. mask-Glascow coma scale total
50. mask-Glascow coma scale verbal response
51. mask-Glucose
52. mask-Heart Rate
53. mask-Height
54. mask-Mean blood pressure
55. mask-Oxygen saturation
56. mask-Respiratory rate
57. mask-Systolic blood pressure
58. mask-Temperature
59. mask-Weight
60. mask-pH

**List of 25 patient disorders involved in Phenotyping in MIMIC-III dataset**

1. Acute and unspecified renal failure
2. Acute cerebrovascular disease
3. Acute myocardial infarction
4. Cardiac dysrhythmias
5. Chronic kidney disease
6. Chronic obstructive pulmonary disease
7. Complications of surgical/medical care
8. Conduction disorders
9. Congestive heart failure; non hypertensive
10. Coronary atherosclerosis and related
11. Diabetes mellitus with complications
12. Diabetes mellitus without complication
13. Disorders of lipid metabolism
14. Essential hypertension
15. Fluid and electrolyte disorders
16. Gastrointestinal haemorrhage
17. Hypertension with complications
18. Other liver diseases
19. Other lower respiratory disease
20. Other upper respiratory disease
21. Pleurisy; pneumothorax; pulmonary collapse
22. Pneumonia
23. Respiratory failure; insufficiency; arrest
24. Septicemia (except in labour)
25. Shock

# D  IMPLEMENTATION DETAILS

## D.1  Model Architectures

**NHS and eICU datasets.** For tabular datasets, we use a multi-layer perceptron (MLP) (Rumelhart et al., 1986) with a single hidden layer of $h$ units, ReLU activation, and a sigmoid output layer. Dropout (0.25) is applied after the hidden layer. We set $h = 256$ for eICU and $h = 64$ for the NHS datasets.

To assess cross-architecture generalisation on eICU, we consider additional MLP variants summarised in Table 6, which vary in width and depth while keeping activation functions and regularisation fixed.

Table 6: MLP architectures for NHS and eICU datasets.

| Model | Architecture | Key Details |
|---|---|---|
| MLP-1 | Wider MLP | $2h$ units, ReLU, dropout 0.25 |
| MLP-2 | Wider MLP | $4h$ units, ReLU, dropout 0.25 |
| MLP-3 | Deeper MLP | 2 layers $(h, 2h)$, ReLU, dropout 0.25 |
| MLP-4 | Deeper MLP | 3 layers $(h, 2h, 4h)$, ReLU, dropout 0.25 |

**MIMIC-III datasets.** For time-series data, we use a temporal convolutional network (TCN) (Bai et al., 2018) as the backbone for dataset condensation. The model consists of a single residual temporal block with 64 channels, kernel size 9, dilation 1, BatchNorm, PReLU activations, and dropout (0.75). The network processes a $48 \times 60$ multivariate time series, with temporal features mean-pooled and passed through a linear output layer. For in-hospital mortality prediction, the output dimension is 1; for phenotyping, the architecture is unchanged except that the output dimension is 25.

For cross-architecture evaluation, we consider additional TCN variants and LSTM baselines summarised in Table 7. The TCN variants extend the base model through multi-scale convolutions and increased depth, while the LSTM models provide a complementary recurrent architecture for comparison.

Table 7: Cross-architecture evaluation models on MIMIC-III.

| Model | Architecture | Key Details |
|---|---|---|
| TCN-1 | Multi-scale TCN | 1 block, kernels [3,5,7], 192 channels, dropout 0.5 |
| TCN-2 | Multi-scale TCN | 2 blocks, kernels [3,5], 256 channels/layer, dropout 0.5, exp. dilation |
| LSTM-1 | LSTM | 1 layer, hidden dim 128, unidirectional, dropout 0.25 |
| LSTM-2 | LSTM | 1 layer, hidden dim 256, unidirectional, dropout 0.25 |

## D.2  Teacher Trajectories

For each dataset, we generate 50 teacher trajectories by training the backbone networks from independent random initialisations.

**MTT trajectories.** For MTT, trajectories are generated using the SGD optimiser, with dataset-specific optimisation settings summarised in Table 8.

**FTD trajectories.** For FTD, trajectories are generated using Generalised Sharpness-Aware Minimisation (GSAM). A linear learning rate schedule is used, decaying to zero over $t_{\max}$ optimisation steps. The perturbation radius $\rho$ is coupled to the learning rate and decreases linearly from 1 to 0 over training. Unless otherwise specified, GSAM uses the same base hyperparameters as SGD.

**DATM trajectories.** For DATM, GSAM-based trajectories are used for the tabular datasets (eICU, Oxford, Portsmouth, Birmingham), following the same configuration as FTD. For MIMIC-III (IHM and Phenotyping), we instead use standard SGD trajectories, as GSAM-based trajectories consistently underperform MTT in this structured time-series setting.

Table 8: SGD (MTT) teacher trajectory optimisation settings.

| Dataset | LR | Momentum | Epochs |
|---|---|---|---|
| MIMIC-III (IHM) | 0.02 | 0.0 | 60 |
| MIMIC-III (Phenotyping) | 0.05 | 0.9 | 60 |
| eICU | 0.02 | 0.9 | 100 |
| Oxford | 0.02 | 0.9 | 100 |
| Portsmouth | 0.01 | 0.9 | 100 |
| Birmingham | 0.02 | 0.9 | 100 |

**Linear and convexified trajectories.** For MCT, we learn convexified trajectories from the MTT teacher trajectories using two learned anchor points per trajectory, following the original method. For the convexified linear variant, we retain the same $\beta$-projection scheme but restrict the trajectory to a single segment between the initial and final parameters of the teacher trajectory. For the linear trajectory ablation, we use the straight line connecting the initial and final parameters of the corresponding SGD trajectory, without anchor points or $\beta$-projection.

**Bézier trajectories.** For BTM, each full teacher trajectory is replaced by a single quadratic Bézier surrogate connecting its initialisation $\boldsymbol{\theta}_0$ and final state $\boldsymbol{\theta}_T$, together with a learnable control point $\boldsymbol{\phi}$. The continuous segments used during condensation are sampled from this surrogate, as described in Section 5. The optimisation of $\boldsymbol{\phi}$ is described below.

### D.3   Control Point Optimisation

The control point $\boldsymbol{\phi}$ is initialised at the midpoint of the trajectory endpoints and optimised to minimise the average training loss along the Bézier curve (Algorithm 2). Since one endpoint corresponds to a random initialisation, this objective does not enforce uniformly low loss across the entire path; instead, it discourages unnecessary excursions and promotes a smoother trajectory with lower average loss between endpoints.

---

**Algorithm 2** Control Point Optimisation

---

**Require:** SGD trajectory endpoints $\boldsymbol{\theta}_0, \boldsymbol{\theta}_T$, dataset $\mathcal{D}$, learning rate $\eta_\phi$, convergence tolerance $\epsilon$, maximum iterations $T_{\max}$, Monte Carlo samples $N_{MC}$
**Ensure:** Optimised control point $\boldsymbol{\phi}^*$
1: Initialise $\boldsymbol{\phi} \leftarrow \frac{\boldsymbol{\theta}_0 + \boldsymbol{\theta}_T}{2}$
2: $t \leftarrow 0$
3: **while** $t < T_{\max}$ **do**
4:      Sample $\{t_i\}_{i=1}^{N_{MC}} \sim \mathcal{U}(0,1)$
5:      $\mathcal{L}_{\mathrm{avg}} \leftarrow 0$
6:      **for** $i = 1$ to $N_{MC}$ **do**
7:          $\boldsymbol{\theta}_{t_i} \leftarrow (1-t_i)^2 \boldsymbol{\theta}_0 + 2t_i(1-t_i)\boldsymbol{\phi} + t_i^2 \boldsymbol{\theta}_T$
8:          Sample mini-batch $\mathcal{B} \subset \mathcal{D}$
9:          $\mathcal{L}_{\mathrm{avg}} \leftarrow \mathcal{L}_{\mathrm{avg}} + \frac{1}{N_{MC}} \mathcal{L}_{\mathrm{CE}}(f_{\boldsymbol{\theta}_{t_i}}, \mathcal{B})$
10:     **end for**
11:     $\boldsymbol{g} \leftarrow \nabla_\phi \mathcal{L}_{\mathrm{avg}}$
12:     **if** $\|\boldsymbol{g}\|_2 < \epsilon$ **then**
13:        **break**
14:     **end if**
15:     $\boldsymbol{\phi} \leftarrow \boldsymbol{\phi} - \eta_\phi \boldsymbol{g}$
16:     $t \leftarrow t + 1$
17: **end while**
18: **return** $\boldsymbol{\phi}^* = \boldsymbol{\phi}$

---

Gradients are computed via automatic differentiation:

$$\nabla_{\boldsymbol{\phi}}\mathcal{L}_{\text{avg}} = \frac{1}{N_{MC}}\sum_{i=1}^{N_{MC}} \nabla_{\boldsymbol{\theta}}\mathcal{L}_{\text{CE}}(f_{\boldsymbol{\theta}_{t_i}}, \mathcal{B}) \cdot \frac{\partial \boldsymbol{\theta}_{t_i}}{\partial \boldsymbol{\phi}}, \tag{73}$$

where $\frac{\partial \boldsymbol{\theta}_{t_i}}{\partial \boldsymbol{\phi}} = 2t_i(1 - t_i)$.

**Hyperparameters.** We set $\eta_{\phi} = 10^{-2}$, $\epsilon = 10^{-5}$, $T_{\max} = 300$, and $N_{MC} = 5$.

**Computational cost.** Control point optimisation requires approximately 1–10 equivalent training epochs per trajectory, depending on dataset size. Specifically, we use 2 epochs for Portsmouth and eICU, 5 for Oxford and Birmingham, and 8 for the MIMIC-III datasets. This cost is incurred once during trajectory construction and amortised over all subsequent condensation iterations.

## D.4 Dataset Condensation

**BTM hyperparameters.** Trajectory parameters are selected via validation. We use continuous segments of length $\Delta t = 0.2$, with $t_s \sim \mathcal{U}(0, 0.8)$ and $t_e = t_s + 0.2$. The student performs $N = 30$ inner-loop updates with learning rate $\eta_s = 0.01$. Synthetic data is optimised using SGD with meta learning rate $\eta_x = 100$ and momentum 0.9, while $\eta_s$ is jointly meta-optimised (learning rate $10^{-4}$, momentum 0.5). The batch size is set to $b = \max(2 \times ipc, 256)$, and condensation runs for $T_{\max} = 20{,}000$ iterations.

**Baseline configurations.** MTT and FTD use discrete trajectory segments of length $M = 5$ epochs with $N = 60$ student updates, and share the same synthetic data optimisation settings as BTM.

DATM also uses $M = 5$ and $N = 60$, but samples segments from an expanding trajectory interval. Let $(T^-, T, T^+)$ denote the lower bound, current upper bound, and final upper bound of the sampling range. Segments are initially drawn from $[T^-, T]$ and progressively expanded to $[T^-, T^+]$, aligning trajectory difficulty with the synthetic budget.

For eICU and NHS datasets:

$$(T^-, T, T^+) = \begin{cases} (0, 20, 40), & 50 \ ipc, \\ (10, 20, 60), & 100 \ ipc, \\ (20, 40, 100), & 200, 500 \ ipc. \end{cases}$$

For MIMIC-III (IHM and Phenotyping):

$$(T^-, T, T^+) = \begin{cases} (0, 10, 25), & 50 \ ipc, \\ (5, 15, 40), & 100 \ ipc, \\ (20, 40, 50), & 200 \ ipc, \\ (30, 40, 60), & 500 \ ipc. \end{cases}$$

For path complexity ablations (MCT and linear variants), all methods use the same segment length $\Delta t$ and number of inner-loop steps $N$ as BTM.

## D.5 Evaluation Protocol

**Model selection during condensation.** Synthetic datasets are evaluated every 10 outer iterations by training randomly initialised models from scratch, and selected based on validation AUPRC. We found AUPRC to be a more reliable selection metric than AUROC, as high AUPRC consistently corresponded to strong AUROC, but not vice versa. Evaluation settings during condensation are summarised in Table 9.

**Final evaluation.** Final results are obtained by retraining models from scratch on the selected synthetic datasets using longer training schedules (Table 10).

Table 9: Evaluation settings during condensation.

| Dataset | LR | Momentum | Epochs |
|---|---|---|---|
| MIMIC-III (IHM) | 0.02 | 0.9 | 60 |
| MIMIC-III (Phenotyping) | 0.05 | 0.9 | 60 |
| eICU, Oxford, Portsmouth, Birmingham | 0.05 | 0.9 | 50 |

Table 10: Final evaluation settings.

| Dataset | LR | Momentum | Epochs |
|---|---|---|---|
| MIMIC-III (IHM) | 0.02 | 0.9 | 80 |
| MIMIC-III (Phenotyping) | 0.05 | 0.9 | 80 |
| eICU, Oxford, Portsmouth, Birmingham | 0.05 | 0.9 | 100 |

**Cross-architecture evaluation.** We evaluate generalisation using alternative architectures. For eICU, we use the MLP variants described in Section D.1. For MIMIC-III, we evaluate TCN and LSTM variants. Within each model family, we use the same optimisation settings as the corresponding base architecture. For LSTM models on MIMIC-III, we instead use a learning rate of 0.01, momentum 0.9, and 30 epochs.

## E  Additional Experiments

### E.1  Inner Loop Steps

The inner loop controls the number of gradient updates the student model takes when trained on the synthetic data before matching against the expert trajectory.

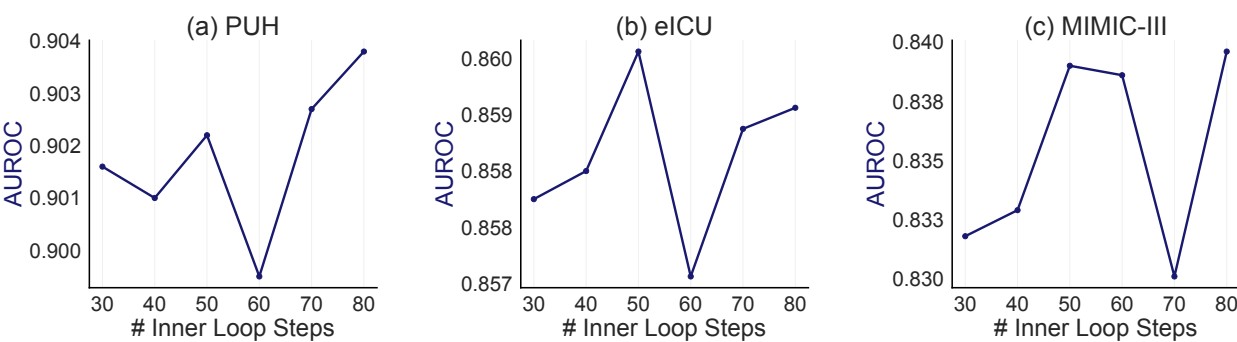

Figure 7: Impact of inner-loop steps $N$ on AUROC performance at 200 *ipc*. BTM achieves strong performance with only 30 steps, reducing computational overhead. Similar trends observed for AUPRC.

In standard TM, this is tied to the expert optimisation epochs M, but with Bézier surrogates the continuous parameterisation $t \in [0, 1]$ decouples segment length ($t_{\text{start}}$, $t_{\text{end}}$) from discrete optimisation steps. As a result, the optimal number of steps must be determined empirically. Figure 7 ablates $N$ at 200 *ipc*. While prior methods typically require $N \geq 40$ (Guo et al., 2024), BTM sustains strong AUROC and AUPRC even at $N = 30$, demonstrating that Bézier surrogates provide stable supervision, enabling the student to train effectively with few gradient steps.

### E.2  Initialisation Strategy for the Synthetic Dataset

Table 11 examines the impact of synthetic-data initialisation on condensation performance in eICU. We compare *real* initialisation, which seeds synthetic inputs from real training samples, with *random* initialisation,

Table 11: Initialisation strategy comparison on eICU. Random initialisation remains competitive while providing stronger privacy guarantees.

| | Init. | IPC | | | |
|---|---|---|---|---|---|
| | | **50** | **100** | **200** | **500** |
| **AUROC** | Real | $0.854_{\pm 0.002}$ | $0.859_{\pm 0.001}$ | $0.861_{\pm 0.001}$ | $0.874_{\pm 0.002}$ |
| | Random | $0.852_{\pm 0.009}$ | $0.858_{\pm 0.004}$ | $0.853_{\pm 0.004}$ | $0.862_{\pm 0.002}$ |
| **AUPRC** | Real | $0.479_{\pm 0.002}$ | $0.486_{\pm 0.001}$ | $0.476_{\pm 0.001}$ | $0.506_{\pm 0.001}$ |
| | Random | $0.463_{\pm 0.015}$ | $0.474_{\pm 0.007}$ | $0.475_{\pm 0.008}$ | $0.499_{\pm 0.004}$ |

which samples inputs from class-conditional Gaussian distributions. The two strategies yield broadly comparable performance, although real initialisation provides a modest advantage at 500 *ipc*. These results suggest that BTM is not strongly dependent on direct seeding from real examples, and that competitive condensation can still be achieved from a purely synthetic starting point.

This comparison is important because the initialisation scheme affects how directly the optimisation begins from the observed data distribution. Real initialisation can place synthetic inputs closer to the data manifold at the outset and may therefore improve optimisation, whereas random initialisation provides a weaker direct dependence on individual training examples at initialisation. However, formal privacy guarantees do not follow from the initialisation strategy alone. Rather, they arise from applying differential privacy to the condensation pipeline itself. Under such a mechanism, both real and random initialisation can be used within a privacy-preserving framework, while random initialisation may still offer a conceptually cleaner starting point with reduced direct dependence on observed samples.

### E.3 Teacher Segment Length in BTM

Table 12 summarises the effect of teacher segment length, $\Delta t = t_e - t_s$, on the Portsmouth dataset. When $\Delta t$ is too small, the induced target displacement is overly restrictive; when $\Delta t$ is too large, the corresponding displacement exceeds what the student can reliably realise within its optimisation budget. We find that $\Delta t = 0.2$ offers the best balance between these two effects, and accordingly use this value in all experiments, as discussed earlier.

Table 12: Impact of continuous segment length ($\Delta t$) on AUPRC for the Portsmouth dataset. A segment length of 0.2 provides the optimal look-ahead window for trajectory alignment.

| Segment Length ($\Delta t$) | IPC | | |
|---|---|---|---|
| | **100** | **200** | **500** |
| 0.1 | $0.558_{\pm 0.012}$ | $0.582_{\pm 0.007}$ | $0.589_{\pm 0.007}$ |
| **0.2** | $\mathbf{0.572}_{\pm 0.001}$ | $\mathbf{0.596}_{\pm 0.001}$ | $\mathbf{0.609}_{\pm 0.001}$ |
| 0.3 | $0.564_{\pm 0.010}$ | $0.585_{\pm 0.008}$ | $0.593_{\pm 0.005}$ |
| 0.4 | $0.556_{\pm 0.011}$ | $0.579_{\pm 0.009}$ | $0.581_{\pm 0.004}$ |

