# OpenReview forum: "Geometric Characterisation and Structured Trajectory Surrogates for Clinical Dataset Condensation"
_TMLR — Under review for TMLR_

### Review · Reviewer_y5V2 · 2026-05-26

**Summary Of Contributions:**

The paper introduces a geometric analysis of trajectory matching (TM) for dataset condensation, demonstrating that a fixed synthetic dataset can only reproduce teacher parameter updates that fall within its reachable gradient span. This creates a conditional representability bottleneck when the teacher's supervision signal's spectrum is broad. To address this, the authors propose Bézier Trajectory Matching (BTM), which replaces jagged, high-rank SGD trajectories with optimized, low-rank quadratic Bézier paths between initial and final model states. The model is evaluated across five clinical datasets spanning tabular and time-series modalities, and it shows a competitive performance, particularly in precision-recall while reducing trajectory storage requirements by one order of magnitude.

Key strengths in this paper are the clear geometric motivation, simple method, strong clinical evaluation on imbalanced real-world cohorts, and substantial storage savings. However, the paper is hindered by key weaknesses, including the gap between the theory and its empirical validation, numerical irregularities in the main tables, limited evidence for the claimed reachable-span mechanism, and privacy wording that should be made more precise.

**Audience:**

Yes

**Audience Explanation:**

The paper would be of interest to researchers working on dataset condensation, trajectory matching, efficient training, and clinical machine learning. The idea of replacing stochastic teacher trajectories with structured low-dimensional Bézier surrogates is simple and potentially useful. The clinical setting also makes the storage and data-governance motivation practically relevant, while the geometric framing may be applicable beyond healthcare.

**Broader Impact Concerns:**

The paper includes a broader impact statement and correctly notes that BTM does not provide formal privacy guarantees. However, the authors should ensure this distinction is stated consistently throughout the paper, especially in the discussion of random initialization. The paper should also more explicitly discuss the risk that low-rank compression may filter out rare or subgroup-specific clinical signals, potentially worsening downstream performance for underrepresented patient groups.

**Claims And Evidence:**

Yes

**Claims Explanation:**

The empirical claim that BTM provides competitive utility for condensed clinical datasets is mostly supported by the reported results, particularly AUPRC improvements in low-prevalence settings such as Birmingham. The reported trajectory storage reduction is also clearly supported. There are table irregularities that require correction. In Table 3, the Random baseline AUROC values and standard deviations appear identical for eICU and MIMIC-III across all ipc values, despite these being distinct datasets and input modalities. In Table 2(c), the Random baseline at ipc=500 reports an unusually large variance relative to the other reported standard deviations and may be a typo.

**Requested Changes:**

- Fix table inconsistencies. Audit the duplicate Random baseline entries in Table 3 and clarify or correct the anomalous standard deviations in Table 2(c) and Table 3(a).
- Empirically validate the reachable-span mechanism.
- Clarify baseline adaptation and tuning. Provide more detail on how MCT and other trajectory-matching baselines were adapted to tabular and time-series clinical data, and whether all methods received comparable validation and hyperparameter tuning budgets.
- Correct privacy phrasing. Revise statements suggesting that random initialization provides “stronger privacy guarantees.” Random initialization may reduce direct dependence on individual examples, but it does not provide formal privacy guarantees without a mechanism such as differential privacy.
- Add subgroup, rare-label, or per-condition analyses to assess whether low-rank condensation preserves rare clinical signals.
- Provide sensitivity studies for Bézier-specific choices such as number of surrogates M, control-point optimization budget, Monte Carlo samples in Algorithm 2, and segment length.
- Improve reproducibility details, especially for the public eICU and MIMIC-III experiments, including code, seeds, preprocessing scripts, and hyperparameter search ranges.
- Clean up minor presentation issues, including typos, repeated appendix table numbering, and unclear captions.

---

### Review · Reviewer_inzj · 2026-06-11

**Summary Of Contributions:**

This work considers the problem of distilling a dataset into a smaller, synthetic dataset such that models trained on the smaller dataset will perform similarly to as if they were trained on the larger dataset. Most works consider a trajectory matching approach, where one aims to find a dataset for which trajectories of SGD will behave similarly to when the full dataset is available. This work argues that such approaches do not perform well when the spectrum of the teacher displacement matrix is large, since the synthetic dataset is constrained by the span of the student gradients (which is typically small). The authors propose to analyze teacher trajectories given by quadratic Bezier curves between the initial iterate, final iterate, and a suitably chosen control point. This ensures that the teacher trajectories live in a low-dimensional subspace, allowing better fits by the student trajectories. The authors compare their approach on several clinical datasets and show that the Bezier trajectory matching (BTM) approach demonstrates solid performance.

**Strengths**
- The authors provide simple, intuitive theory to support why an alternative form of trajectory matching would be useful and how BTM addresses this.
- The method is also storage-efficient.
- Compared to other trajectory-based approaches, the method performs at least as well, if not better than other approaches.

**Weaknesses**
- Some aspects of the theory should be empirically verified or strengthened in terms of their preciseness (see changes).
- While the clinical focus is interesting, it would be nice to empirically justify some of the claims that are made about why the emphasis is on the clinical applications. See below.

**Audience:**

Yes

**Audience Explanation:**

Researchers working in dataset distillation would be interested in the findings of this paper, along with those who use clinical datasets as the main application of this work is in this area.

**Claims And Evidence:**

Yes

**Claims Explanation:**

The paper has a nice balance of simple theoretical motivation for their method along with a solid amount of experimental comparisons. Certain aspects of the theory could be fleshed out empirically (discussed more below).

**Requested Changes:**

The following changes would strengthen the work in my view:
- In Theorem 4, the quantities $\kappa$ and $D$ determine the size of the bound. It would be useful to see empirically whether these quantities are indeed small, especially $D$.
- As discussed on page 11 in the Experimental Setup, the authors motivate why clinical datasets are a particularly good setting to analyze trajectory-matching approaches. It would be useful to empirically verify some of these claims, especially the claim that clinical datasets suffer from this "wide spectrum". Although Figure 2 provides some of this evidence, it would strengthen the paper to measure the actual projection residuals induced by learned synthetic datasets, e.g., e.g., $||(I-P_G)A||_F^2$ after constructing the learned synthetic dataset such as in Theorem 2. I think this would strengthen the emphasis on clinical datasets.
- Some of the theory is more-or-less very standard results in linear algebra and can be shortened by citing such standard facts. For example, the formula for the orthogonal projection $\arg \min_{v \in V} ||x - v||^2 = P_Vx$ is entirely re-derived in equations (7)-(15) in the appendix.
- While it is good that the authors compared to other trajectory matching approaches, I would have expected comparisons to non-trajectory-based approaches such as distribution matching or gradient matching.